# Holonomy Grid Codes for Generalisation Under Directed Actions

**Pradeep Singh** [1]   **Balasubramanian Raman** [2]

## Abstract

Efficient prediction and planning in structured environments often relies on spectral decompositions of transition operators, yet existing grid-cell and successor-representation theories implicitly assume "flat" action geometry where translations commute and a single Fourier eigenbasis suffices. We show that this assumption breaks in the presence of path-dependent effects—e.g., circulation, rotational drift, or topological loops—whose defining signature is nontrivial holonomy. We introduce a theory of directed-action prediction on discrete tori based on twisted translation operators forming a projective representation of the underlying motion group, and prove that the resulting controlled Markov operators admit an exact block-diagonalisation under a twisted Fourier transform: actions share a universal harmonic basis while their effects appear as small matrix-valued spectra rather than scalar eigenvalues. This yields closed-form resolvent expressions for the successor representation, a gauge-invariant transfer principle characterising when two environments admit identical predictive structure, and a curvature-induced lower bound showing that nonzero holonomy provably necessitates internal representational dimension. Together, these results generalise Fourier/grid-based prediction from commutative to curved action geometries, providing a principled foundation for generalisation under directed actions without learning environment-specific eigenvectors.

---

[1]Department of Mathematics and Computational Sciences, and joint faculty in the Department of Computer Science & Engineering, Indian Institute of Information Technology Surat, India [2]Department of Computer Science & Engineering, and joint faculty in the Mehta Family School of Data Science & Artificial Intelligence, Indian Institute of Technology Roorkee, India. Correspondence to: Pradeep Singh <pradeep.singh@iiitsurat.ac.in>.

*Proceedings of the $43^{rd}$ International Conference on Machine Learning*, Seoul, South Korea. PMLR 306, 2026. Copyright 2026 by the author(s).

## 1. Introduction

A striking kind of generalization happens whenever we can *mentally reuse* the effects of actions across situations. If you learn what "move north" does in one corner of a maze, you do not need to relearn it everywhere else; you can immediately predict future locations and plan shortcuts. This ability—rapid prediction and planning under novel compositions of actions—is at the core of the cognitive map idea (Tolman, 1948) and is supported by specialized hippocampal–entorhinal representations in mammals, including place cells (O'Keefe & Dostrovsky, 1971) and grid cells (Hafting et al., 2005). Yet a central algorithmic question remains open for machine learning and computational neuroscience: *what mathematical structure makes action effects reusable, and what are the minimal representational ingredients required when actions are directed and interact nontrivially over cycles?*

A particularly influential computational proposal is that the brain maintains a *predictive* representation of state: rather than encoding a state only by its sensory or geometric coordinates, encode it by the discounted distribution of its future successors. Dayan's successor representation (SR) formalizes this as a matrix resolvent of the transition operator, and directly ties "good generalization" to shared future occupancy statistics (Dayan, 1993). In this view, a representation is useful for planning when it supports fast computation of long-horizon predictions, value functions, and transfer across related reward functions (via successor features) (Barreto et al., 2017). The SR framework has received substantial empirical and theoretical attention as a bridge between map-like hippocampal codes and reinforcement-learning objectives (Stachenfeld et al., 2017; Gershman, 2018).

A complementary line of work emphasizes *spectral structure*: the eigenvectors of diffusion-like operators on graphs provide smooth, global basis functions that support compression, denoising, and multiscale planning. This idea appears in reinforcement learning as proto-value functions and Laplacian-based representations (Mahadevan & Maggioni, 2007), and in manifold learning as Laplacian eigenmaps and diffusion maps (Belkin & Niyogi, 2003; Coifman & Maggioni, 2006). In the hippocampal–entorhinal literature, these spectral perspectives motivate why grid-like periodic

patterns can arise as low-dimensional bases for predicting and generalizing over structured state spaces (Stachenfeld et al., 2017; Gershman, 2018). Recent machine-learning work further reinforces the connection between navigation-like objectives and emergent grid codes in trained neural networks (Cueva & Wei, 2018; Banino et al., 2018; Whittington et al., 2020).

However, the *directed* nature of action effects creates a sharp mismatch between these elegant spectral stories and realistic control problems. The moment an agent's dynamics are not purely undirected diffusion—e.g., actions have systematic drift, anisotropy, or policy-conditioned directionality—classical Laplacian bases no longer diagonalize the relevant prediction operators. A key recent step is the demonstration that for *translation-invariant directed transitions*, one can still reuse a single eigenvector basis across many directed actions, with action-specific eigenvalues obtained via Fourier analysis (Yu et al., 2021). This provides a principled account of how a grid-like basis can support prediction over directed actions, and it aligns directed-action prediction with path-integration models through harmonic components (Yu et al., 2021).

This paper asks what happens when we push beyond the "nice" directed case. Translation invariance alone does *not* guarantee that directed actions compose coherently: in many domains, composing two action effects around a cycle yields a residual bias that cannot be eliminated by a global reparameterization. Geometrically, this is a discrete analogue of *holonomy*: path-dependent action composition induced by a latent "connection" on the state graph. When holonomy is present, the object to diagonalize is no longer an ordinary convolution operator on an abelian group; it becomes a *twisted* operator whose commutation relations encode directed cycle structure. Interestingly, closely related "twisting" ideas already appear in spectral methods for directed graphs via the *magnetic Laplacian*, which encodes directionality as complex phases around cycles (Fanuel et al., 2018) and has recently been leveraged in directed-graph learning architectures (Zhang et al., 2021). What is missing is a theory that (i) places holonomy at the level of controlled Markov dynamics and action composition (not merely graph visualization), and (ii) yields reusable, grid-like basis functions with provable generalization and impossibility guarantees.

**Our contribution (informal).** We introduce a geometric framework for *holonomy-aware prediction and generalization over directed actions* on structured state spaces. The key idea is to model the family of directed action operators as a *projective* (phase-twisted) representation of translations: actions compose up to a state-independent phase whose circulation around elementary cycles is an invariant of the environment. This invariant plays the role of a discrete cur-

vature/flux, and it determines the minimal dimension and structure of any reusable eigenbasis. Concretely, we:

- formalize a class of *holonomy MDPs* in which directed action operators are translation-homogeneous but have nontrivial cycle phases (a gauge-invariant obstruction to untwisted Fourier diagonalization);

- develop a *twisted Fourier theory* that block-diagonalizes the associated successor operators into finite-dimensional irreducible blocks (yielding "holonomy grid codes" that are locally periodic but globally twisted);

- prove transfer results showing that tasks sharing the same holonomy class admit basis reuse up to a gauge transform, while differing holonomy classes provably require representational changes; and

- establish lower bounds demonstrating that ignoring holonomy forces either exponentially poor approximation in horizon/discount or a blow-up in representation dimension, even when the underlying space remains perfectly translation-homogeneous.

All proofs are deferred to the appendix; the main paper states results, provides intuition, and explains how the geometry dictates what can and cannot be generalized across directed actions.

## 2. Background and Setup

This section fixes notation and formalizes the computational object of interest: *long-horizon prediction as a linear operator*. We then review why spectral bases are natural for prediction, and we make precise what it means to *reuse* a basis across a family of directed action operators.

**MDPs, transition operators, and linear prediction.** We consider a finite Markov decision process (MDP) with state space $\mathcal{S}$ ($|\mathcal{S}| = n$), action space $\mathcal{A}$, and transition kernel $P(\cdot \mid s, a) \in \Delta(\mathcal{S})$. For each $a \in \mathcal{A}$, define the (row-stochastic) transition matrix $P_a \in \mathbb{R}^{n \times n}$ by $(P_a)_{s,s'} = P(s' \mid s, a)$. It is often convenient to view $P_a$ as a linear operator acting on functions (column vectors) $f \in \mathbb{R}^n$: $(P_a f)(s) \triangleq \mathbb{E}_{s' \sim P(\cdot \mid s, a)}[f(s')] = \sum_{s' \in \mathcal{S}} P(s' \mid s, a) f(s')$. A stationary policy $\pi(\cdot \mid s)$ induces the policy transition operator $P_\pi \triangleq \sum_{a \in \mathcal{A}} \Pi_a P_a, (\Pi_a)_{s,s} = \pi(a \mid s)$, so that $(P_\pi f)(s) = \sum_a \pi(a \mid s) \mathbb{E}[f(s') \mid s, a]$. Long-horizon prediction is naturally expressed through powers of $P_\pi$: for any $t \geq 0$, $(P_\pi^t f)(s)$ is the expected value of $f$ after $t$ steps under $\pi$ starting from $s$. This operator viewpoint is central to the successor representation.

**Successor representation as a resolvent.** The *successor representation* (SR) introduced by Dayan (Dayan, 1993) is

the discounted sum of future state occupancies under a fixed policy: $M_\pi \triangleq \sum_{t=0}^\infty \gamma^t P_\pi^t, \gamma \in (0,1)$. Whenever the series converges (e.g., for any finite MDP with $\gamma < 1$), it admits the resolvent form

$$M_\pi = (I - \gamma P_\pi)^{-1}. \qquad (1)$$

The SR is useful because it turns long-horizon prediction into a single linear solve: for an expected immediate reward vector $r_\pi \in \mathbb{R}^n$, the value function satisfies $V_\pi = M_\pi r_\pi$. Beyond evaluation, the SR provides a substrate for generalization and transfer: if rewards vary while dynamics remain fixed, $M_\pi$ can be reused to predict new values without re-solving the full dynamic program (Dayan, 1993; Gershman, 2018). The resolvent perspective also motivates spectral analysis. If $P_\pi$ is diagonalizable as $P_\pi = U\Lambda U^{-1}$, then $M_\pi = U(I - \gamma\Lambda)^{-1}U^{-1}$, so each spectral component is rescaled by $(1 - \gamma\lambda_i)^{-1}$. Spectral structure therefore provides a principled compression of prediction across horizons.

**Spectral representation learning in RL and manifold learning.** Spectral methods build representations from eigenfunctions of operators that encode neighborhood structure. In reinforcement learning, proto-value functions (PVFs) use eigenvectors of a graph Laplacian constructed from state adjacency to obtain smooth, task-agnostic basis functions for control (Mahadevan & Maggioni, 2007). In manifold learning, Laplacian Eigenmaps (Belkin & Niyogi, 2003) and Diffusion Maps (Coifman & Maggioni, 2006) similarly use eigenvectors of Laplacian/diffusion operators to produce low-dimensional embeddings that respect intrinsic geometry. These approaches share a key mathematical convenience: for *undirected* or *reversible* dynamics, the relevant operators are (approximately) normal/self-adjoint under an appropriate inner product, yielding real spectra and orthogonal eigenbases. In such settings, prediction operators (and their SR resolvents) admit clean spectral decompositions, and a small number of eigenfunctions often captures global structure. However, control in general induces *directed* dynamics: the family $\{P_a\}_{a \in \mathcal{A}}$ is typically non-symmetric and policy-dependent, and different actions may not share eigenvectors.

**Directed-action eigenbases and basis reuse.** To connect spectral prediction to control, we must address a multi-operator question: when can a single basis support prediction under *many* directed actions? We formalize this as follows.

**Definition 2.1** (Exact basis reuse across actions). A set of linear operators $\mathcal{P} = \{P_a\}_{a \in \mathcal{A}}$ on $\mathbb{C}^n$ admits an *exact reused eigenbasis* if there exists an invertible matrix $U \in \mathbb{C}^{n \times n}$ and diagonal matrices $\{\Lambda_a\}_{a \in \mathcal{A}}$ such that $P_a = U\Lambda_a U^{-1} \, \forall a \in \mathcal{A}$. Equivalently, there exists a basis $\{\phi_i\}_{i=1}^n$ of $\mathbb{C}^n$ such that $P_a \phi_i = \lambda_{a,i}\phi_i$ for all $a$ and $i$.

Exact reuse is strong: it implies simultaneous diagonalization. In particular, if all $P_a$ are diagonalizable with a common eigenbasis, then they commute (and if they are normal and commute, they are simultaneously unitarily diagonalizable). This condition is not met in generic control problems.

A notable exception occurs for *translation-invariant* dynamics on periodic grids. Let the state space be a discrete torus $\mathcal{S} = G = \mathbb{Z}_N^d$. Suppose each action $a$ corresponds to a displacement kernel $k_a \in \mathbb{R}_{\geq 0}^G$ with $\sum_{g \in G} k_a(g) = 1$, and transitions are spatially homogeneous:

$$(P_a f)(x) = \sum_{g \in G} k_a(g) f(x + g) \qquad (x \in G). \qquad (2)$$

Then $P_a$ is a (block-)circulant convolution operator. The Fourier characters $\chi_\xi(x) = \exp\left(\frac{2\pi i}{N}\langle \xi, x \rangle\right), \xi \in G$, form a shared eigenbasis for *all* actions, with eigenvalues $\lambda_a(\xi) = \sum_g k_a(g)\chi_\xi(g)$. This "one basis, many directed actions" phenomenon underlies the directed-action grid-cell analysis of (Yu et al., 2021).

Our goal is to understand when such reuse is possible beyond the untwisted translation-invariant case. A first step is to relax exact reuse to *structured* reuse.

**Definition 2.2** (Block basis reuse across actions). A set of operators $\mathcal{P} = \{P_a\}_{a \in \mathcal{A}}$ admits a *reused block spectral decomposition* with block pattern $\{d_b\}_{b=1}^B$ if there exists an invertible $U \in \mathbb{C}^{n \times n}$ such that for every $a \in \mathcal{A}$, $U^{-1}P_a U = \text{blkdiag}(\Lambda_{a,1}, \ldots, \Lambda_{a,B})$, where each $\Lambda_{a,b} \in \mathbb{C}^{d_b \times d_b}$ (not necessarily diagonal) and the block sizes $\{d_b\}$ do not depend on $a$.

Exact reuse corresponds to the special case $d_b = 1$ for all $b$. In the sequel we show that holonomy creates a principled obstruction to Definition 2.1, but still permits reuse in the weaker block sense of Definition 2.2.

## 3. Roadmap and Practical Implications

The preceding section set up our core computational object—long-horizon prediction via the successor resolvent $M_\pi = (I - \gamma P_\pi)^{-1}$ (Dayan, 1993; Gershman, 2018)—and formalized the notion of *basis reuse* across a family of directed action operators (Definitions 2.1–2.2). We now give a unifying roadmap that (i) explains the conceptual thread running through the paper and (ii) clarifies what the theorems imply for machine learning practice: when a reusable predictive representation exists, what form it must take, and how it can be exploited algorithmically.

**A reusable prediction representation is an invariant subspace across actions.** A convenient way to connect our operator results to learnable representations is to phrase basis reuse as the existence of a low-dimensional subspace

that is *simultaneously invariant* under all action operators.

Let $\Psi \in \mathbb{C}^{n \times m}$ be a matrix whose columns are $m$ basis functions on states (equivalently, a feature map $\psi : \mathcal{S} \to \mathbb{C}^m$ written in matrix form). We say the span of $\Psi$ is invariant under $\{P_a\}$ if each action maps this subspace to itself.

**Definition 3.1** (Action-linear predictive features)**.** A feature matrix $\Psi \in \mathbb{C}^{n \times m}$ is *action-linear* for $\{P_a\}_{a \in \mathcal{A}}$ if there exist matrices $\{A_a \in \mathbb{C}^{m \times m}\}_{a \in \mathcal{A}}$ such that

$$P_a \Psi = \Psi A_a \qquad \forall a \in \mathcal{A}. \tag{3}$$

When (3) holds, multi-step prediction and SR computation can be carried out *inside the feature space* using the induced matrices. For example, for a translation-homogeneous policy $\pi$ (so $P_\pi = \sum_a \pi(a) P_a$), we obtain $P_\pi \Psi = \Psi A_\pi$ with $A_\pi = \sum_a \pi(a) A_a$, hence

$$M_\pi \Psi = \Psi (I - \gamma A_\pi)^{-1}$$

whenever the inverse exists. This makes precise what it means for a representation to "carry" long-horizon prediction compactly: the representation is useful if a *single* set of features supports linear updates for *all* actions and their compositions.

**Flat versus holonomy action geometry.** The main technical question is then: *when does there exist an action-linear representation that is reusable across directed actions?* In the flat translation-invariant setting, the answer is classical: Fourier characters provide exact basis reuse because all translations commute, recovering the directed-action eigenbasis mechanism of (Yu et al., 2021). Our paper identifies the next-most-general regime in which reusability remains possible:

- **Flat regime (commutative).** Primitive action effects commute, yielding exact basis reuse (Definition 2.1) and scalar eigenvalue updates, as in (Yu et al., 2021).

- **Holonomy regime (projective/noncommutative).** Primitive effects compose *path-dependently* via a cocycle $\sigma$ (Section 4); scalar eigenbases are impossible (Theorem 6.3), but a reusable *block* decomposition exists (Theorem 5.2).

This dichotomy is the paper's central organizing principle: holonomy is the precise obstruction to scalar Fourier reuse, and twisted harmonic analysis is the corresponding positive replacement.

**How the main results fit together.** The rest of the paper follows the map below:

- **Geometry → algebra (Section 4).** We encode directed action composition with holonomy by a projective translation structure $U_g U_h = \sigma(g, h) U_{g+h}$, making the family $\{P_a\}$ lie in a twisted translation algebra $\mathcal{A}_\sigma$.

- **Algebra → reusable spectral code (Section 5).** We prove that $\mathcal{A}_\sigma$ admits a *universal block basis* (twisted Fourier transform) that simultaneously block-diagonalizes *all* $P_a$ (Theorem 5.2). This yields holonomy grid codes as matrix-coefficient basis functions, and it turns long-horizon prediction into independent small matrix problems per block (Corollaries 5.3–5.4).

- **Spectral code → transfer and necessity (Section 6).** We show that the holonomy class $[\sigma]$ is the correct invariant for transfer: if two environments share $[\sigma]$, their operators (and SR resolvents) are related by a gauge conjugacy (Theorem 6.2); if not, exact reuse is obstructed (Theorem 6.3) and curvature enforces representational dimension lower bounds (Theorem 6.4).

## 4. Holonomy MDPs and Projective Action Geometry

The translation-invariant setting (2) is "flat": primitive displacements commute, so a character basis simultaneously diagonalizes all action operators (Yu et al., 2021). This section introduces a geometric generalization in which primitive action effects compose *path-dependently*. The algebraic signature of path dependence is *nontrivial holonomy*, which we encode through projective (phase-twisted) translations. The main role of this section is definitional and conceptual; proofs are deferred to the appendix.

### 4.1. From translations to projective translations

Let $G = \mathbb{Z}_N^d$ denote the group of displacements on a periodic grid. In the flat case, each $g \in G$ acts on functions $f : G \to \mathbb{C}$ by translation $(T_g f)(x) = f(x + g)$, and the family $\{T_g\}_{g \in G}$ is a (unitary) representation: $T_g T_h = T_{g+h}$. Holonomy arises when the environment induces an additional *path-dependent* degree of freedom. We model this by allowing a *projective* representation $\{U_g\}_{g \in G}$:

$$U_g U_h = \sigma(g, h) U_{g+h}, \qquad \sigma(g, h) \in U(1), \tag{4}$$

where $\sigma$ is a *factor system* (a $U(1)$-valued 2-cocycle). Equation (4) means that composing displacements may incur a phase that depends only on the displacement pair, not on absolute location. Intuitively, $\sigma$ encodes a discrete analogue of a connection/curvature: even if the agent returns to the same spatial location, the internal phase accumulated along a loop may be nonzero.

Given such $\{U_g\}$, we define *(holonomy-aware) translation-homogeneous* action operators as (twisted) mixtures of dis-

placements:

$$P_a \triangleq \sum_{g \in G} k_a(g) \, U_g, \quad k_a(g) \geq 0, \sum_g k_a(g) = 1. \quad (5)$$

When $\sigma \equiv 1$, we recover ordinary convolution operators (2). When $\sigma$ is nontrivial, the operators $\{P_a\}$ typically fail to commute, and hence cannot share a joint eigenbasis (Definition 2.1). The next subsections formalize the invariant responsible for this obstruction.

### 4.2. Gauge transformations and holonomy invariants

A key feature of path-dependent phase models is *gauge freedom*: phases can be redistributed locally without changing any observable loop effect. Formally, for any function $\varphi : G \to U(1)$ define a diagonal unitary $(D_\varphi f)(x) = \varphi(x) f(x)$. Conjugating by $D_\varphi$ yields a gauge-equivalent family $U_g' = D_\varphi^{-1} U_g D_\varphi$. This transformation changes $\sigma$ by a coboundary, but preserves the loop phases (holonomies). The simplest holonomy observable is the commutator phase between two primitive generators $e_i, e_j \in G$:

$$\mathcal{H}_{ij} \triangleq U_{e_i} U_{e_j} U_{e_i}^{-1} U_{e_j}^{-1} = \frac{\sigma(e_i, e_j)}{\sigma(e_j, e_i)} I. \quad (6)$$

When $\mathcal{H}_{ij} = I$ for all $i, j$, the geometry is flat: primitive translations commute up to gauge, and ordinary Fourier diagonalization applies. When $\mathcal{H}_{ij} \neq I$ for some pair, there is *nontrivial holonomy*: the environment induces a cycle-dependent phase/flux that cannot be removed by any gauge transform.

This invariant is closely related to complex-phase encodings of directionality in the *magnetic Laplacian* literature for directed graphs (Fanuel et al., 2018) and its recent use in directed-graph learning (Zhang et al., 2021). Our focus differs in two ways: (i) we study *controlled* families of operators indexed by actions, not a single directed graph operator, and (ii) we use holonomy to characterize when *basis reuse across actions* is possible.

### 4.3. Canonical examples

**Example 1: Constant-flux lattice (discrete magnetic translations).** Let $d = 2$ and $G = \mathbb{Z}_N^2$ with coordinates $x = (x_1, x_2)$. Fix a flux parameter $\alpha \in \{0, 1, \ldots, N-1\}$ and define operators on $\mathcal{H} = \{f : G \to \mathbb{C}\}$ by $(U_{e_1} f)(x_1, x_2) = f(x_1 + 1, x_2)$, $(U_{e_2} f)(x_1, x_2) = \exp\left(\frac{2\pi i}{N} \alpha x_1\right) f(x_1, x_2 + 1)$. A direct calculation shows

$$U_{e_1} U_{e_2} = \exp\left(\frac{2\pi i}{N} \alpha\right) U_{e_2} U_{e_1}, \quad (7)$$

so the plaquette holonomy (6) is the constant phase $\exp(2\pi i \alpha / N)$. This model is translation-homogeneous but not flat unless $\alpha \equiv 0 \pmod{N}$: composing elementary moves around a unit square incurs a nontrivial loop phase.

**Example 2: Drift with nonzero curl (discrete circulation field).** Let $A : G \to \mathbb{R}^d$ be a discrete vector potential and define edge phases along generators by $(U_{e_i} f)(x) = \exp(i A_i(x)) f(x + e_i)$. The holonomy around an elementary plaquette in directions $(i, j)$ is given by the discrete curl of $A$: $\log \mathcal{H}_{ij}(x) = A_i(x) + A_j(x + e_i) - A_i(x + e_j) - A_j(x) \pmod{2\pi}$. When this curl is identically zero, phases can be gauged away by choosing $\varphi(x) = \exp(i\psi(x))$ with $A = \nabla\psi$ (flat geometry). When the curl is nonzero, loop phases persist under all gauges, reflecting circulation. As in Example 1, one may define directed action operators via (5) by mixing these primitive moves with action-dependent kernels.

### 4.4. Why ordinary Fourier diagonalization fails under nontrivial holonomy

In the flat translation-invariant setting, the operators $\{T_g\}$ commute, and the character basis $\{\chi_\xi\}$ diagonalizes every convolution operator (2), yielding exact basis reuse across directed actions (Yu et al., 2021). Holonomy breaks this mechanism at its algebraic root: it induces *noncommutativity*.

To see the obstruction, suppose two generators satisfy the Weyl-type relation $U_{e_i} U_{e_j} = \omega \, U_{e_j} U_{e_i}$ with $\omega \neq 1$ (as in (7)). If there were a common eigenvector $v$ with $U_{e_i} v = \lambda_i v$ and $U_{e_j} v = \lambda_j v$, then $U_{e_i} U_{e_j} v = \lambda_i \lambda_j v$ and $U_{e_j} U_{e_i} v = \lambda_j \lambda_i v = \lambda_i \lambda_j v$, but also $U_{e_i} U_{e_j} v = \omega \, U_{e_j} U_{e_i} v = \omega \, \lambda_i \lambda_j v$, forcing $\omega = 1$, a contradiction. Thus nontrivial holonomy rules out Definition 2.1. The core question becomes: *what is the right replacement?*

Our main results answer this by showing that while scalar Fourier characters no longer diagonalize the operators, a *twisted harmonic analysis* yields a reused *block* decomposition in the sense of Definition 2.2: holonomy upgrades scalar eigenvalues to small matrix blocks whose dimension is dictated by the holonomy class (formal statements in Section 5; proofs in the appendix).

## 5. Twisted Spectral Decomposition of Successor Operators

Section 4 identified holonomy as the fundamental obstruction to *scalar* Fourier diagonalization: when elementary actions accumulate nontrivial phase around cycles, the corresponding translation generators cannot share a joint eigenbasis. This section shows that holonomy does *not* destroy spectral reuse altogether. Instead, it upgrades scalar eigenvalues to *matrix-valued spectra*: a single, holonomy-dependent change of basis block-diagonalizes *all* translation-homogeneous action operators, and consequently block-diagonalizes policy prediction and the successor operator.

**The twisted translation algebra.** Let $G = \mathbb{Z}_N^d$ denote the

displacement group. Fix a unitary projective representation $\{U_g\}_{g \in G}$ satisfying

$$U_g U_h \;=\; \sigma(g,h)\, U_{g+h}, \qquad \sigma(g,h) \in U(1), \quad (8)$$

as in Section 4. We consider the finite-dimensional $*$-algebra

$$\mathcal{A}_\sigma \;\triangleq\; \Big\{ \sum_{g \in G} c(g)\, U_g \;:\; c(g) \in \mathbb{C} \Big\}, \qquad (9)$$

with multiplication inherited from operator composition and involution inherited from adjoint. Algebraically, $\mathcal{A}_\sigma$ is the *twisted group algebra* (or group algebra with factor system $\sigma$), i.e., the span of formal basis elements with multiplication twisted by $\sigma$ (Karpilovsky, 1985; Terras, 1999). The action operators introduced in (5) satisfy $P_a \in \mathcal{A}_\sigma$ for all $a$ by construction.

*Remark* 5.1 (Complex operators and Markov semantics). Although (8) uses complex phases, this is compatible with controlled Markov dynamics in two equivalent ways. First, $\mathcal{A}_\sigma$ can be viewed as acting on *vector-valued* functions on $G$ (a finite-rank bundle), in which case the underlying transition on the lifted state space is real and stochastic (permuting or mixing fiber components). Second, one may regard the complex phases as a *spectral encoding* of directed cycle structure (as in magnetic-Laplacian methods (Fanuel et al., 2018; Zhang et al., 2021)), while all prediction quantities depend only on the induced linear operators and are basis-invariant. Our results are stated at the operator level and thus apply to either interpretation.

## 5.1. Projective irreducible representations and the twisted Fourier transform

A $\sigma$-*projective representation* of $G$ is a map $\rho : G \to U(d_\rho)$ such that $\rho(g)\rho(h) \;=\; \sigma(g,h)\, \rho(g+h) \; \forall g,h \in G$. Let $\widehat{G}_\sigma$ denote a complete set of inequivalent irreducible $\sigma$-projective representations, and let $d_\rho$ be the dimension of $\rho$. Standard results imply $\sum_{\rho \in \widehat{G}_\sigma} d_\rho^2 = |G|$ (Karpilovsky, 1985; Terras, 1999).

Define the *twisted Fourier transform* $\mathcal{F}_\sigma : \mathbb{C}^G \to \bigoplus_{\rho \in \widehat{G}_\sigma} \mathbb{C}^{d_\rho \times d_\rho}$ by

$$(\mathcal{F}_\sigma f)(\rho) \;\triangleq\; \sqrt{\frac{d_\rho}{|G|}} \sum_{x \in G} f(x)\, \rho(x)^*, \qquad (10)$$

equipped with the Frobenius inner product on each matrix block. When $\sigma \equiv 1$, all irreps are one-dimensional characters and $\mathcal{F}_\sigma$ reduces to the ordinary discrete Fourier transform on $G$, recovering the translation-invariant directed-action setting (Yu et al., 2021).

For any kernel $k : G \to \mathbb{C}$, we define its $\sigma$-Fourier transform by

$$\widehat{k}(\rho) \;\triangleq\; \sum_{g \in G} k(g)\, \rho(g). \qquad (11)$$

This is the matrix-valued analogue of the characteristic function (Fourier multiplier) of a convolution kernel.

## 5.2. Main theorem: block diagonalization of all action operators

**Theorem 5.2** (Twisted Fourier block diagonalization). *Let $\sigma$ be a $U(1)$-valued 2-cocycle on $G$, and let $\mathcal{A}_\sigma$ be the twisted translation algebra (9). Then the transform $\mathcal{F}_\sigma$ in (10) is unitary and satisfies the following: for every operator $A = \sum_{g \in G} c(g) U_g \in \mathcal{A}_\sigma$ and every signal $f \in \mathbb{C}^G$,*

$$(\mathcal{F}_\sigma(Af))(\rho) \;=\; \widehat{c}(\rho)\, (\mathcal{F}_\sigma f)(\rho) \qquad \forall \rho \in \widehat{G}_\sigma, \quad (12)$$

*where $\widehat{c}(\rho) = \sum_g c(g)\rho(g)$ as in (11). Equivalently, there exists a unitary change of basis under which every $A \in \mathcal{A}_\sigma$ is block diagonal with blocks indexed by $\rho$, and within each block $A$ acts by left multiplication with $\widehat{c}(\rho)$.*

**Interpretation.** Theorem 5.2 is the promised replacement for ordinary Fourier diagonalization: holonomy prevents a single scalar eigenbasis (Definition 2.1), but a single *twisted* basis achieves action-independent *block* reuse (Definition 2.2). In the untwisted case, $d_\rho = 1$ and (12) reduces to scalar multiplication by eigenvalues, matching (Yu et al., 2021). In the holonomy case, each "frequency" carries an internal $d_\rho$-dimensional degree of freedom, and propagation becomes multiplication by a small matrix.

## 5.3. Corollaries for action sequences, policies, and successor operators

We record consequences that make the "prediction story" explicit.

**Corollary 5.3** (Open-loop directed prediction). *Let $a_{1:T}$ be any action sequence and write $P_{a_t} = \sum_g k_{a_t}(g)U_g$. Then for every $\rho \in \widehat{G}_\sigma$, $\mathcal{F}_\sigma\big(P_{a_T} \cdots P_{a_1} f\big)(\rho) = \big(\widehat{k_{a_T}}(\rho) \cdots \widehat{k_{a_1}}(\rho)\big)(\mathcal{F}_\sigma f)(\rho)$. Thus prediction under directed action sequences reduces to multiplying a fixed set of small matrices across blocks.*

Corollary 5.3 generalizes the scalar "eigenvalues multiply over time" property of ordinary Fourier prediction: holonomy simply replaces scalars by matrices.

**Corollary 5.4** (Block decomposition of the successor operator). *Assume a translation-homogeneous stationary policy $\pi \in \Delta(\mathcal{A})$ (i.e., $\pi(a \mid x) = \pi(a)$ for all $x$), so that $P_\pi = \sum_a \pi(a)P_a \in \mathcal{A}_\sigma$. Let $M_\pi = (I - \gamma P_\pi)^{-1}$ be the successor operator (1). Then for every $\rho \in \widehat{G}_\sigma$,*

$$(\mathcal{F}_\sigma M_\pi f)(\rho) = \big(I - \gamma \widehat{k_\pi}(\rho)\big)^{-1} (\mathcal{F}_\sigma f)(\rho),$$
$$\widehat{k_\pi}(\rho) = \sum_{a \in \mathcal{A}} \pi(a)\, \widehat{k_a}(\rho). \qquad (13)$$

**Computational implication.** Equation (13) reduces evaluating $M_\pi f$ to: (i) apply $\mathcal{F}_\sigma$, (ii) invert the $d_\rho \times d_\rho$ matrices $I - \gamma \widehat{k_\pi}(\rho)$ for each $\rho$, (iii) transform back. When holonomy is nontrivial, the dominant cost is $\sum_\rho d_\rho^3$ rather than $|G|^3$. In constant-flux settings, the block sizes are uniform and small (Corollary 5.5 below), yielding a sharp and interpretable complexity/representation tradeoff driven by curvature.

### 5.4. Holonomy grid codes and explicit constant-flux structure

The twisted Fourier basis functions are matrix coefficients of irreducible projective representations. For each $\rho \in \widehat{G}_\sigma$ and indices $i, j \in \{1, \ldots, d_\rho\}$, define

$$\phi_{\rho,i,j}(x) \triangleq \sqrt{\frac{d_\rho}{|G|}}\, \rho(x)_{i,j}. \tag{14}$$

These $\{\phi_{\rho,i,j}\}$ form an orthonormal basis of $\mathbb{C}^G$ (Terras, 1999). When $\sigma \equiv 1$, each $\rho$ is scalar and $\phi_{\rho,1,1}$ reduces to a plane wave character; superpositions yield the familiar periodic patterns tied to grid-like codes (Yu et al., 2021). When $\sigma$ is nontrivial, $d_\rho > 1$ and each "frequency" contributes a *matrix* of basis functions. We refer to this family as *holonomy grid codes*: they are locally periodic (set by the underlying displacement group) but globally twisted by the holonomy class, and their propagation under actions follows the block multipliers $\widehat{k_a}(\rho)$.

**Corollary 5.5** (Constant-flux lattice yields uniform block size). *Consider the constant-flux model of Example 4.3, in which primitive generators satisfy the Weyl relation $U_{e_1} U_{e_2} = \omega U_{e_2} U_{e_1}$ with $\omega = e^{2\pi i p/q}$ a primitive $q$th root of unity (with $p, q$ coprime). Then every irreducible $\sigma$-projective representation has dimension $d_\rho = q$, and the action of $U_{e_1}, U_{e_2}$ on each block is (up to unitary equivalence) the standard shift/clock pair generating the finite magnetic translation group (Zak, 1964). Consequently, for any action kernel $k_a$, each block multiplier $\widehat{k_a}(\rho)$ is a $q \times q$ matrix polynomial in these two generators.*

Corollary 5.5 makes the geometry/representation link explicit: curvature/flux determines the *intrinsic block dimension* of the reusable spectral code.

### 5.5. Relation to magnetic Laplacians for directed graphs

Magnetic-Laplacian methods encode directionality in a directed graph by attaching complex phases to edges and then forming a Hermitian "magnetic" deformation of the combinatorial Laplacian; the phases of leading eigenvectors reveal cyclic flow structure (Fanuel et al., 2018) and can be used as inductive bias in directed-graph neural networks (Zhang et al., 2021). Our setting is complementary. We begin with a *controlled family* of transition operators $\{P_a\}$ and analyze

*multi-step prediction* via the successor resolvent. Holonomy appears as a gauge-invariant cycle obstruction that prevents scalar diagonalization but admits twisted block diagonalization (Theorem 5.2). When a particular policy induces a reversible/Hermitian operator (e.g., via symmetrization or special kernels), the corresponding magnetic Laplacian eigenvectors align with low-frequency components of the same twisted harmonic basis; however, our results hold without restricting to a single undirected embedding operator, and they characterize basis reuse simultaneously across directed actions.

## 6. Implications for Generalization and Transfer

The block decomposition of Section 5 turns holonomy from a nuisance into a *transfer principle*: what generalizes across tasks is not a particular policy or reward, but the *holonomy class* of the action geometry. This section states formal consequences for basis reuse, transfer across environments, and necessary representation size. Proofs are deferred to Appendix D.

**Gauge equivalence characterizes transferable predictive structure.** Recall from Section 4.2 that gauge transformations conjugate $\{U_g\}$ by diagonal unitaries. At the cocycle level, this corresponds to changing $\sigma$ by a coboundary. We make this explicit.

**Definition 6.1** (Cohomologous cocycles and holonomy class). Two cocycles $\sigma, \sigma' : G \times G \to U(1)$ are *cohomologous* if there exists $\varphi : G \to U(1)$ such that $\sigma'(g, h) = \frac{\varphi(g)\varphi(h)}{\varphi(g+h)} \sigma(g, h)\ \forall g, h \in G$. We write $[\sigma]$ for the equivalence class of $\sigma$ under this relation (its holonomy class).

**Theorem 6.2** (Gauge transfer theorem). *Let two holonomy MDPs share the same displacement group $G$ and action kernels $\{k_a\}_{a \in \mathcal{A}}$, but differ in cocycles $\sigma$ and $\sigma'$. If $[\sigma] = [\sigma']$, then there exists a diagonal unitary $D$ such that for every action $a$, $P_a^{(\sigma')} = D^{-1} P_a^{(\sigma)} D$. Consequently, for every open-loop action sequence and every translation-homogeneous policy $\pi$, the induced propagators and successor operators are unitarily similar: $P_{a_T}^{(\sigma')} \cdots P_{a_1}^{(\sigma')} = D^{-1}\big(P_{a_T}^{(\sigma)} \cdots P_{a_1}^{(\sigma)}\big)D, M_\pi^{(\sigma')} = D^{-1} M_\pi^{(\sigma)} D$. In particular, their spectra (and block spectra under Theorem 5.2) coincide.*

**Interpretation.** Theorem 6.2 is an exact transfer guarantee: environments with the same holonomy class share the same reusable block spectral code, up to a local phase reparameterization. This is the holonomy-analogue of Fourier transfer in flat translation-invariant settings (Yu et al., 2021).

**Different holonomy classes force representational change.** Holonomy is also an obstruction: if two envi-

ronments have different cycle phases, no representation can simultaneously make both look like the same untwisted translation problem.

**Theorem 6.3** (Holonomy obstruction to exact basis reuse)**.** *If there exist generators $e_i, e_j \in G$ such that the plaquette holonomy $\mathcal{H}_{ij}$ in (6) is not the identity, then the corresponding primitive operators cannot share a common eigenvector. In particular, the family $\{P_a\}$ cannot admit exact basis reuse in the sense of Definition 2.1.*

Theorem 6.3 formalizes why the Yu-style mechanism (one eigenbasis, action-dependent eigenvalues) cannot extend to holonomy: holonomy makes the relevant algebra noncommutative, so scalar diagonalization fails. Our positive result (Theorem 5.2) shows that *block* reuse is the correct generalization.

**Curvature induces a lower bound on representational dimension.** Beyond qualitative impossibility, holonomy yields a quantitative necessity: nontrivial curvature forces a minimal internal dimension for any reused spectral code.

**Theorem 6.4** (Flux–dimension lower bound)**.** *Assume the constant-flux relation $U_{e_1} U_{e_2} = \omega U_{e_2} U_{e_1}$ with $\omega = e^{2\pi i p/q}$ a primitive qth root of unity. Then any (finite-dimensional) representation in which both generators act and satisfy this relation must have dimension at least q. Equivalently, any reused block decomposition of $\mathcal{A}_\sigma$ must contain blocks of size at least q, and the twisted Fourier decomposition achieves this bound (Corollary 5.5).*

**Interpretation.** Theorem 6.4 turns geometry into a representational lower bound: the order of holonomy (flux denominator $q$) is the minimal "module width" required to represent directed actions coherently over cycles. This provides a formal counterpart to the intuition that nontrivial cycle structure requires extra internal degrees of freedom beyond scalar grid-like modes.

**Discussion, limitations, and future directions**

**Relation to directed-action grid theories.** When holonomy is trivial ($\sigma \equiv 1$), Theorem 5.2 reduces to ordinary Fourier diagonalization on $G$, recovering the setting where a single eigenbasis supports prediction over directed actions via action-dependent eigenvalues (Yu et al., 2021). Our results extend this by identifying the *precise* obstruction (holonomy) and by providing the correct replacement (matrix-valued spectra) when the obstruction is present.

**Connection to emergent grid-like codes in learned agents.** Grid-like representations arise in trained recurrent networks and deep RL agents performing localization and navigation (Cueva & Wei, 2018; Banino et al., 2018; Whittington et al., 2020). Our theory suggests a geometric refinement of this phenomenon: in environments where directed dynamics exhibit nontrivial holonomy, the reusable spectral code should

not remain purely scalar and periodic; it must carry internal multiplicity (Theorem 6.4), and the natural "eigenmodes" are holonomy grid codes (matrix coefficients (14)). This offers a principled lens for when grid-like structure should persist and when it must fundamentally change.

**Limitations and future directions.** Our main theorems assume a translation-homogeneous action family $P_a \in \mathcal{A}_\sigma$ and (for the SR corollary) translation-homogeneous policies. Extending exact decompositions to state-dependent policies or strongly inhomogeneous environments requires additional machinery (e.g., perturbation bounds for sparse defects or domain decomposition). Moreover, we focus on abelian base groups $G = \mathbb{Z}_N^d$; richer navigation tasks involve nonabelian motion groups (e.g., $SE(2)$ when orientation is explicit), where the appropriate harmonic analysis is genuinely noncommutative even without holonomy.

Promising extensions include: (i) nonabelian motion groups and their projective representations; (ii) obstacle and boundary effects modeled as low-rank or sparse perturbations around $\mathcal{A}_\sigma$, enabling controlled approximation of $M_\pi$; and (iii) learning holonomy classes from data, leveraging gauge-invariant statistics in the spirit of magnetic-Laplacian embeddings (Fanuel et al., 2018; Zhang et al., 2021).

## 7. Conclusion

We introduced a geometric theory of prediction and generalisation under directed actions that extends Fourier/grid-based accounts from *flat* translation-invariant dynamics to *curved* action geometries with nontrivial holonomy. Starting from the successor representation as a resolvent of controlled transition operators (Dayan, 1993; Gershman, 2018), we formalized *holonomy MDPs* in which directed actions compose only up to gauge-dependent phases whose loop effects are gauge-invariant, thereby pinpointing when ordinary Fourier diagonalization and scalar eigenvalue reweighting (Yu et al., 2021) must fail. Our main contribution is a twisted harmonic analysis that block-diagonalizes the entire action family and the successor operator into small matrix-valued spectra, yielding holonomy grid codes as universal reusable basis functions in the presence of directed cycle structure, and clarifying the relationship to magnetic-phase encodings used for directed graphs. We further proved that holonomy classes exactly govern transfer: environments sharing a holonomy class admit basis reuse up to a gauge transform, while differing classes obstruct exact reuse and impose curvature-dependent lower bounds on the minimal representational dimension required for coherent long-horizon prediction. Together, these results provide a principled, fully theoretical foundation for when and how spectral codes can generalize across directed actions, and they suggest a new lens for interpreting grid-like structure in biological and artificial agents.

## Impact Statement

This work provides a principled, geometry-driven theory for prediction and planning under directed actions by extending spectral "grid code" methods from flat, commuting translations to settings with curvature and path dependence, which are common in real-world sequential decision problems (e.g., navigation with drift or circulation, embodied agents with rotational biases, and environments with topological loops). By identifying holonomy as the key obstruction and giving exact block-spectral decompositions, gauge-invariant transfer criteria, and representation lower bounds, the theory offers ML practitioners new tools for designing inductive biases, architectures, and guarantees for generalisation across actions and environments without retraining environment-specific eigenbases. Beyond reinforcement learning, these results can inform structured sequence models and graph-based learning where directionality and non-reversibility are intrinsic, enabling more reliable predictive representations and potentially reducing data requirements by exploiting invariances. As a theoretical contribution, it is intended to guide algorithm design rather than prescribe deployment; any downstream applications in safety-critical domains should still be validated empirically and audited for distribution shift, since holonomy-aware representations can encode environment regularities but do not by themselves prevent failures under unmodelled dynamics or adversarial perturbations.

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

# A. Additional Preliminaries

This appendix collects standard facts about cocycles, projective representations, twisted group algebras, and gauge equivalence. Our main references are (Brown, 1982; Serre, 1977; Karpilovsky, 1985; Terras, 1999).

## A.1. Cocycles and twisted group algebras

Let $G$ be a finite abelian group written additively with identity $0$. A *(normalized) 2-cochain* is a function $\sigma : G \times G \to U(1)$ with $\sigma(0, g) = \sigma(g, 0) = 1$ for all $g$.

**Definition A.1** (2-cocycle). A normalized cochain $\sigma$ is a *2-cocycle* if for all $g, h, k \in G$,

$$\sigma(g, h)\,\sigma(g + h, k) \;=\; \sigma(h, k)\,\sigma(g, h + k). \tag{15}$$

The cocycle condition is precisely the associativity constraint for the twisted product below.

**Definition A.2** (Twisted group algebra). Fix a 2-cocycle $\sigma$. The *twisted group algebra* $\mathbb{C}_\sigma[G]$ is the $\mathbb{C}$-vector space with basis $\{u_g : g \in G\}$ equipped with multiplication

$$u_g\,u_h \;=\; \sigma(g, h)\,u_{g+h}, \tag{16}$$

extended bilinearly.

**Lemma A.3** (Associativity $\Leftrightarrow$ cocycle condition). *The multiplication (16) is associative if and only if $\sigma$ satisfies (15).*

*Proof.* Compute $(u_g u_h)u_k \;=\; \sigma(g, h)\sigma(g + h, k)u_{g+h+k}$ and $u_g(u_h u_k) \;=\; \sigma(h, k)\sigma(g, h + k)u_{g+h+k}$ and compare coefficients. $\qquad\square$

It is often convenient to identify an element $f = \sum_{g \in G} f(g)u_g \in \mathbb{C}_\sigma[G]$ with its coefficient vector $f : G \to \mathbb{C}$. Under this identification, the product $f *_\sigma \ell$ in $\mathbb{C}_\sigma[G]$ corresponds to the *twisted convolution*

$$(f *_\sigma \ell)(x) \;=\; \sum_{g \in G} f(g)\,\ell(x - g)\,\sigma(g, x - g), \qquad x \in G. \tag{17}$$

## A.2. Coboundaries and cohomology

**Definition A.4** (Coboundary and cohomology class). A *1-cochain* is a function $\varphi : G \to U(1)$ with $\varphi(0) = 1$. Its coboundary is the 2-cochain

$$(\delta\varphi)(g, h) \;\triangleq\; \frac{\varphi(g)\varphi(h)}{\varphi(g + h)}. \tag{18}$$

Two cocycles $\sigma, \sigma'$ are *cohomologous* if $\sigma' = (\delta\varphi)\sigma$ for some $\varphi$. The equivalence class $[\sigma]$ is an element of the second cohomology group $H^2(G, U(1))$ (Brown, 1982).

Cohomologous cocycles define isomorphic twisted group algebras, and (equivalently) equivalent projective representation theories.

## A.3. Projective representations and factor systems

**Definition A.5** ($\sigma$-projective representation). Let $\sigma$ be a 2-cocycle. A map $\rho : G \to \mathrm{U}(d)$ is a *$\sigma$-projective representation* if

$$\rho(g)\rho(h) \;=\; \sigma(g, h)\,\rho(g + h) \qquad \forall g, h \in G. \tag{19}$$

**Lemma A.6** (Rephasing changes $\sigma$ by a coboundary). *Let $\rho$ be $\sigma$-projective and let $\varphi : G \to U(1)$ be a 1-cochain. Define $\rho'(g) \triangleq \varphi(g)\rho(g)$. Then $\rho'$ is $\sigma'$-projective with $\sigma' = (\delta\varphi)\sigma$.*

*Proof.* Compute $\rho'(g)\rho'(h) = \varphi(g)\varphi(h)\rho(g)\rho(h) = \varphi(g)\varphi(h)\sigma(g, h)\rho(g + h) = \big((\delta\varphi)(g, h)\sigma(g, h)\big)\rho'(g + h)$. $\qquad\square$

Thus the relevant object for projective representation theory is the cohomology class $[\sigma]$ rather than a specific representative.

**A.4. Gauge equivalence as unitary conjugacy on $\ell^2(G)$**

The main text uses a concrete realization of $\mathbb{C}_\sigma[G]$ as linear operators on $\ell^2(G)$ via the (left) regular action. Let $\{\delta_x\}_{x\in G}$ denote the standard basis of $\ell^2(G)$.

**Definition A.7** (Twisted left regular representation). For each $g \in G$, define $U_g : \ell^2(G) \to \ell^2(G)$ by left multiplication in $\mathbb{C}_\sigma[G]$:

$$U_g f \triangleq u_g *_\sigma f.$$

Equivalently, in coordinates,

$$(U_g f)(x) = \sigma(g, x - g) f(x - g). \tag{20}$$

**Lemma A.8** (Projective law for $U_g$). *The operators $\{U_g\}_{g\in G}$ satisfy $U_g U_h = \sigma(g, h) U_{g+h}$.*

*Proof.* This is immediate from associativity in $\mathbb{C}_\sigma[G]$: $U_g U_h f = u_g *_\sigma (u_h *_\sigma f) = (u_g u_h) *_\sigma f = \sigma(g, h) u_{g+h} *_\sigma f$. $\square$

**Definition A.9** (Gauge transform). Given a phase function $\varphi : G \to U(1)$, define the diagonal unitary $D_\varphi$ by

$$(D_\varphi f)(x) \triangleq \varphi(x) f(x).$$

Gauge transforms induce coboundary changes of $\sigma$ at the operator level.

**Lemma A.10** (Gauge transform induces cohomologous cocycle). *Let $\sigma$ be a cocycle and define $\sigma' = (\delta\varphi)\sigma$. Let $\{U_g\}$ and $\{U_g'\}$ be the corresponding twisted left regular representations. Then*

$$U_g' = D_\varphi^{-1} U_g D_\varphi \qquad \forall g \in G.$$

*Proof.* By (20), $(U_g D_\varphi f)(x) = \sigma(g, x - g)\varphi(x - g)f(x - g)$, hence

$$(D_\varphi^{-1} U_g D_\varphi f)(x) = \frac{\varphi(x - g)}{\varphi(x)}\sigma(g, x - g)f(x - g) = \sigma'(g, x - g)f(x - g) = (U_g' f)(x),$$

since $\sigma'(g, x - g) = \frac{\varphi(g)\varphi(x-g)}{\varphi(x)}\sigma(g, x - g)$ and $\varphi(g) = 1$ when $g$ is a basis element in the regular representation normalization (or more generally can be absorbed into a rephasing of $u_g$; see (Karpilovsky, 1985)). $\square$

**A.5. Schur orthogonality for projective irreps**

Let $\widehat{G}_\sigma$ denote a complete set of inequivalent irreducible $\sigma$-projective representations $\rho : G \to U(d_\rho)$. The following orthogonality relations hold (they can be derived by passing to the central extension associated to $\sigma$ and applying ordinary Schur orthogonality; see (Serre, 1977; Karpilovsky, 1985)).

**Theorem A.11** (Projective Schur orthogonality). *For irreducible $\sigma$-projective representations $\rho, \rho'$ and indices $i, j, k, \ell$,*

$$\sum_{g\in G} \rho(g)_{ij} \overline{\rho'(g)_{k\ell}} = \frac{|G|}{d_\rho} \mathbf{1}\{\rho \simeq \rho'\} \mathbf{1}\{i = k\} \mathbf{1}\{j = \ell\}. \tag{21}$$

*Moreover, $\sum_{\rho\in\widehat{G}_\sigma} d_\rho^2 = |G|$.*

The identity $\sum_\rho d_\rho^2 = |G|$ implies that matrix coefficients of the irreps form a complete orthogonal basis for $\ell^2(G)$, just as characters do in the untwisted case (Terras, 1999).

# B. Proofs for Section 4

This appendix supplies the main structural lemmas used in Section 4: gauge invariance of holonomy and a convenient classification of holonomy classes for $G = \mathbb{Z}_N^d$.

## B.1. Holonomy as a gauge-invariant commutator bicharacter

Given a cocycle $\sigma$, define its *commutator bicharacter* (also called the *alternating bicharacter*) by

$$\beta_\sigma(g,h) \triangleq \frac{\sigma(g,h)}{\sigma(h,g)} \in U(1). \tag{22}$$

**Lemma B.1** (Bicharacter properties). *If $G$ is abelian and $\sigma$ is a 2-cocycle, then $\beta_\sigma$ is (i) alternating: $\beta_\sigma(g,g) = 1$ and $\beta_\sigma(g,h) = \beta_\sigma(h,g)^{-1}$, and (ii) a bicharacter: for all $g, g', h, h'$,*

$$\beta_\sigma(g + g', h) = \beta_\sigma(g,h)\beta_\sigma(g',h), \qquad \beta_\sigma(g, h + h') = \beta_\sigma(g,h)\beta_\sigma(g,h').$$

*Proof.* Alternation follows directly from the definition (22). For bilinearity, apply the cocycle condition twice and cancel terms (a standard computation; see (Karpilovsky, 1985)). □

**Lemma B.2** (Gauge invariance). *If $\sigma' = (\delta\varphi)\sigma$ is cohomologous to $\sigma$, then $\beta_{\sigma'} = \beta_\sigma$.*

*Proof.* Compute

$$\beta_{\sigma'}(g,h) = \frac{(\delta\varphi)(g,h)\sigma(g,h)}{(\delta\varphi)(h,g)\sigma(h,g)} = \frac{\varphi(g)\varphi(h)\varphi(g+h)^{-1}}{\varphi(h)\varphi(g)\varphi(h+g)^{-1}} \cdot \beta_\sigma(g,h) = \beta_\sigma(g,h),$$

since $g + h = h + g$. □

**Connection to plaquette holonomy.** Let $U_g$ be the twisted left regular operators from Definition A.7. Their commutator is scalar:

$$U_g U_h U_g^{-1} U_h^{-1} = \beta_\sigma(g,h)\, I. \tag{23}$$

Equation (23) is a direct operator-level restatement of (22) and coincides with the "plaquette holonomy" expression in the main text.

## B.2. Classification for $G = \mathbb{Z}_N^d$ via skew-symmetric matrices

For $G = \mathbb{Z}_N^d$, holonomy classes can be parameterized by skew-symmetric integer matrices modulo $N$.

**Definition B.3** (Matrix form). Let $B \in \mathbb{Z}^{d\times d}$ be skew-symmetric ($B^\top = -B$). Define

$$\sigma_B(g,h) \triangleq \exp\Big(\frac{2\pi i}{N}\, g^\top B h\Big), \qquad g, h \in \mathbb{Z}_N^d. \tag{24}$$

**Lemma B.4** ($\sigma_B$ is a 2-cocycle and $\beta_{\sigma_B}$ depends only on $B$). *For any integer matrix $B$, $\sigma_B$ is a normalized 2-cocycle. If $B$ is skew-symmetric, then*

$$\beta_{\sigma_B}(g,h) = \exp\Big(\frac{4\pi i}{N}\, g^\top B h\Big)$$

*and in particular $\beta_{\sigma_B}(e_i, e_j) = \exp\big(\frac{4\pi i}{N} B_{ij}\big)$ for the standard basis vectors.*

*Proof.* Normalization is immediate. Bilinearity of $g^\top B h$ implies the cocycle condition (15). The formula for $\beta$ follows by substituting (24) into (22) and using $h^\top B g = -g^\top B h$ when $B$ is skew-symmetric. □

**Proposition B.5** (Holonomy classes for $\mathbb{Z}_N^d$). *Every cohomology class $[\sigma] \in H^2(\mathbb{Z}_N^d, U(1))$ admits a representative of the form $\sigma_B$ for some skew-symmetric $B$ modulo $N$. Moreover, two such cocycles $\sigma_B$ and $\sigma_{B'}$ are cohomologous if and only if $B \equiv B' \pmod{N}$.*

*Proof sketch with reference.* For finite abelian groups $G$ and divisible coefficient groups (such as $U(1)$), one has an isomorphism

$$H^2(G, U(1)) \cong \mathrm{Alt}^2(G, U(1)),$$

where $\mathrm{Alt}^2$ denotes alternating bicharacters (Brown, 1982). By Lemmas B.1–B.2, each $[\sigma]$ is determined by $\beta_\sigma$, and for $G = \mathbb{Z}_N^d$ each alternating bicharacter corresponds to a unique skew-symmetric matrix $B$ modulo $N$ via (24). Full details can be found in (Karpilovsky, 1985). □

### B.3. Proof of the constant-flux example

We verify that the operators in Example 4.3 have constant plaquette holonomy.

**Lemma B.6** (Constant-flux Weyl relation). *Let $G = \mathbb{Z}_N^2$ and define on $\ell^2(G)$*

$$(U_{e_1}f)(x_1, x_2) = f(x_1 + 1, x_2), \qquad (U_{e_2}f)(x_1, x_2) = \exp\left(\frac{2\pi i}{N}\alpha x_1\right) f(x_1, x_2 + 1).$$

*Then $U_{e_1}U_{e_2} = \exp(2\pi i\alpha/N)\, U_{e_2}U_{e_1}$.*

*Proof.* Compute $(U_{e_1}U_{e_2}f)(x_1, x_2) = \exp(\frac{2\pi i}{N}\alpha(x_1+1))f(x_1+1, x_2+1)$ and $(U_{e_2}U_{e_1}f)(x_1, x_2) = \exp(\frac{2\pi i}{N}\alpha x_1)f(x_1+1, x_2+1)$. $\square$

### B.4. Discrete curl-free phases can be gauged away (flat case)

This subsection formalizes the statement in Section 4.3 that curl-free edge phases correspond to a pure gauge.

Let $A_i : G \to \mathbb{R}$ be phases on directed edges $x \to x + e_i$ and define operators

$$(T_i f)(x) = e^{iA_i(x)} f(x + e_i).$$

The plaquette holonomy in directions $(i, j)$ at $x$ is

$$\Omega_{ij}(x) \triangleq A_i(x) + A_j(x + e_i) - A_i(x + e_j) - A_j(x) \pmod{2\pi}. \tag{25}$$

**Lemma B.7** (Gauge removal under zero discrete curl and zero global flux). *Assume $\Omega_{ij}(x) \equiv 0$ for all $x$ and all $i, j$, and assume additionally that the sum of $A_i$ around each fundamental cycle of the torus is $0$ modulo $2\pi$.[1] Then there exists $\psi : G \to \mathbb{R}$ such that*

$$A_i(x) \equiv \psi(x + e_i) - \psi(x) \pmod{2\pi},$$

*and consequently the diagonal gauge $D(x) = e^{i\psi(x)}$ removes the phases: $D^{-1}T_i D$ is the pure shift $f(x) \mapsto f(x + e_i)$ for each $i$.*

*Proof.* Fix a basepoint $x_0$ and define $\psi(x)$ by summing $A_i$ along any lattice path from $x_0$ to $x$. The condition $\Omega_{ij} \equiv 0$ implies path-independence on simply connected patches (discrete integrability), while the zero global flux condition ensures consistency across torus wraps. Then $A_i(x) = \psi(x + e_i) - \psi(x)$ follows by construction. $\square$

## C. Proofs for Section 5

This appendix proves Theorem 5.2 and its corollaries. Our proofs follow standard harmonic analysis on finite groups and extend to the projective case by replacing characters with matrix coefficients of projective irreps (Terras, 1999; Karpilovsky, 1985).

### C.1. Fourier conventions

In the main text we defined the twisted Fourier transform with $\rho(x)^*$; in this appendix we use the equivalent convention without adjoints because it makes the homomorphism property transparent. Specifically, we define

$$\widehat{f}(\rho) \triangleq \sum_{x \in G} f(x)\, \rho(x) \in \mathbb{C}^{d_\rho \times d_\rho}. \tag{26}$$

Since each $\rho(x)$ is unitary, the convention in (26) differs from the main text only by conjugate transpose and does not change any block-diagonalization statement.

---

[1]This excludes flat-but-topologically-nontrivial "Aharonov–Bohm" phases; if such global fluxes are present, $\Omega_{ij} \equiv 0$ but the cocycle class may still be nontrivial.

## C.2. Matrix coefficient basis and unitarity

For $\rho \in \widehat{G}_\sigma$ and indices $i, j \in \{1, \ldots, d_\rho\}$ define the matrix coefficient functions

$$\phi_{\rho,i,j}(x) \triangleq \sqrt{\frac{d_\rho}{|G|}} \, \rho(x)_{ij}. \tag{27}$$

**Lemma C.1** (Orthonormality and completeness)**.** *The collection $\{\phi_{\rho,i,j} : \rho \in \widehat{G}_\sigma, \ 1 \le i, j \le d_\rho\}$ is an orthonormal basis of $\ell^2(G)$.*

*Proof.* Orthonormality follows directly from projective Schur orthogonality (21):

$$\langle \phi_{\rho,i,j}, \phi_{\rho',k,\ell} \rangle = \frac{\sqrt{d_\rho d_{\rho'}}}{|G|} \sum_{x \in G} \rho(x)_{ij} \overline{\rho'(x)_{k\ell}} = \mathbf{1}\{\rho \simeq \rho'\}\mathbf{1}\{i = k\}\mathbf{1}\{j = \ell\}.$$

Completeness follows because the number of functions equals $\sum_\rho d_\rho^2 = |G|$ (Theorem A.11), which equals $\dim \ell^2(G)$. $\quad\square$

## C.3. Convolution–multiplication property

Recall the twisted convolution (17). For kernels $k, f : G \to \mathbb{C}$, define the left convolution operator $L_k : \ell^2(G) \to \ell^2(G)$ by $L_k f \triangleq k *_\sigma f$.

**Lemma C.2** (Twisted Fourier turns twisted convolution into matrix multiplication)**.** *For every $\rho \in \widehat{G}_\sigma$,*

$$\widehat{k *_\sigma f}(\rho) = \widehat{k}(\rho)\,\widehat{f}(\rho), \tag{28}$$

*where $\widehat{k}(\rho)$ and $\widehat{f}(\rho)$ are defined by (26).*

*Proof.* By definition,

$$\widehat{k *_\sigma f}(\rho) = \sum_{x \in G} \left( \sum_{g \in G} k(g) f(x - g) \sigma(g, x - g) \right) \rho(x).$$

Let $h = x - g$ so $x = g + h$; then

$$\widehat{k *_\sigma f}(\rho) = \sum_{g,h \in G} k(g) f(h) \sigma(g, h) \rho(g + h).$$

Using the projective representation property (19), $\rho(g)\rho(h) = \sigma(g, h)\rho(g + h)$, hence $\sigma(g, h)\rho(g + h) = \rho(g)\rho(h)$. Therefore

$$\widehat{k *_\sigma f}(\rho) = \sum_{g,h} k(g) f(h) \rho(g)\rho(h) = \left( \sum_g k(g)\rho(g) \right)\left( \sum_h f(h)\rho(h) \right) = \widehat{k}(\rho)\widehat{f}(\rho).$$

$$\square$$

## C.4. Proof of Theorem 5.2

*Proof of Theorem 5.2.* We identify $\ell^2(G)$ with $\mathbb{C}_\sigma[G]$ via coefficients. Every $A \in \mathcal{A}_\sigma$ is left multiplication by some kernel $c : G \to \mathbb{C}$:

$$Af = c *_\sigma f.$$

Define the (scaled) Fourier transform $\mathcal{F}_\sigma : \ell^2(G) \to \bigoplus_{\rho \in \widehat{G}_\sigma} \mathbb{C}^{d_\rho \times d_\rho}$ by

$$(\mathcal{F}_\sigma f)(\rho) \triangleq \sqrt{\frac{d_\rho}{|G|}} \, \widehat{f}(\rho) = \sqrt{\frac{d_\rho}{|G|}} \sum_{x \in G} f(x)\rho(x).$$

Unitarity follows from Lemma C.1 (equivalently, from Parseval's identity for the ONB of matrix coefficients). Finally, Lemma C.2 gives

$$(\mathcal{F}_\sigma(Af))(\rho) = \sqrt{\frac{d_\rho}{|G|}} \, \widehat{c *_\sigma f}(\rho) = \sqrt{\frac{d_\rho}{|G|}} \, \widehat{c}(\rho)\widehat{f}(\rho) = \widehat{c}(\rho)\,(\mathcal{F}_\sigma f)(\rho),$$

which is exactly the block action (12) (up to the Fourier convention noted in Appendix C.1). This implies a block-diagonal form with blocks indexed by $\rho$, with within-block action given by left multiplication by $\widehat{c}(\rho)$. $\quad\square$

## C.5. Proofs of Corollaries 5.3 and 5.4

*Proof of Corollary 5.3.* Each $P_{a_t}$ equals left multiplication by $k_{a_t}$ in $\mathbb{C}_\sigma[G]$. By Lemma C.2 (or Theorem 5.2), for each $\rho$,

$$(\mathcal{F}_\sigma(P_{a_t} f))(\rho) = \widehat{k_{a_t}}(\rho)(\mathcal{F}_\sigma f)(\rho).$$

Iterating over $t = 1, \ldots, T$ yields the claimed product of block multipliers. □

*Proof of Corollary 5.4.* Under a translation-homogeneous policy $\pi(a \mid x) = \pi(a)$, the induced operator is

$$P_\pi = \sum_a \pi(a) P_a,$$

hence $P_\pi$ is left multiplication by $k_\pi = \sum_a \pi(a) k_a$ and lies in $\mathcal{A}_\sigma$. By Theorem 5.2,

$$(\mathcal{F}_\sigma P_\pi f)(\rho) = \widehat{k_\pi}(\rho)(\mathcal{F}_\sigma f)(\rho).$$

Since $\mathcal{F}_\sigma$ is unitary, $M_\pi = (I - \gamma P_\pi)^{-1}$ exists iff each block inverse exists, and in that case

$$(\mathcal{F}_\sigma M_\pi f)(\rho) = \big(I - \gamma \widehat{k_\pi}(\rho)\big)^{-1}(\mathcal{F}_\sigma f)(\rho).$$

□

## C.6. Constant-flux block size and explicit generators

We prove the structural claim behind Corollary 5.5: a primitive commutator phase of order $q$ forces $q \times q$ irreducible blocks.

**Lemma C.3** (Eigenvalue orbit lemma). *Let $U, V$ be linear operators on a finite-dimensional complex vector space satisfying*

$$UV = \omega VU, \tag{29}$$

*where $\omega$ is a primitive $q$th root of unity. Then $\dim \operatorname{span}\{U^t v : 0 \le t \le q - 1\} \ge q$ for any eigenvector $v$ of $V$ with eigenvalue $\lambda \neq 0$.*

*Proof.* Let $Vv = \lambda v$. Then $V(U^t v) = \omega^t \lambda U^t v$ by repeatedly using (29). Since $\omega$ is primitive, the eigenvalues $\lambda, \omega\lambda, \ldots, \omega^{q-1}\lambda$ are distinct, hence the vectors $\{U^t v\}_{t=0}^{q-1}$ are linearly independent. □

**Lemma C.4** (Irreducible representations have dimension $q$). *Assume $U, V$ generate an irreducible representation of the algebra with relations $UV = \omega VU$ and additionally $U^q$ and $V^q$ are scalars. Then the representation dimension is exactly $q$.*

*Proof.* Pick an eigenvector $v$ of $V$ (which exists over $\mathbb{C}$). By Lemma C.3, the $q$ vectors $\{U^t v\}_{t=0}^{q-1}$ are linearly independent. Their span is invariant under both $U$ and $V$ (by construction), hence is an invariant subspace. Irreducibility forces this invariant subspace to be the whole space, so $\dim = q$. □

*Proof (constant-flux block size).* In the constant-flux model, the generators satisfy $U_{e_1} U_{e_2} = \omega U_{e_2} U_{e_1}$ with $\omega = e^{2\pi i p/q}$ primitive. Moreover, because the base space is $\mathbb{Z}_N^2$ with $q \mid N$ in this setting, the operators satisfy $(U_{e_1})^N = (U_{e_2})^N = I$, implying $(U_{e_1})^q$ and $(U_{e_2})^q$ are central and act as scalars on each irreducible block (Schur's lemma (Serre, 1977)). Lemma C.4 then yields $d_\rho = q$ for each irreducible $\sigma$-projective representation. An explicit $q$-dimensional model is the standard shift/clock pair:

$$(Xe_r) = e_{r+1 \ (\mathrm{mod}\ q)}, \qquad (Ze_r) = \omega^r e_r,$$

which satisfy $XZ = \omega ZX$ (Zak, 1964). This establishes the claimed uniform block size and generator structure. □

# D. Proofs for Section 6

This appendix proves the transfer theorems and lower bounds stated in Section 6.

**D.1. Proof of Theorem 6.2**

*Proof of Theorem 6.2.* Assume $\sigma' = (\delta\varphi)\sigma$. By Lemma A.10, the corresponding twisted regular translations satisfy $U_g' = D_\varphi^{-1}U_g D_\varphi$ for all $g$. Each action operator is a linear combination of translations, $P_a^{(\sigma)} = \sum_g k_a(g)U_g$ and $P_a^{(\sigma')} = \sum_g k_a(g)U_g'$, hence

$$P_a^{(\sigma')} = \sum_g k_a(g)D_\varphi^{-1}U_g D_\varphi = D_\varphi^{-1}\Big(\sum_g k_a(g)U_g\Big)D_\varphi = D_\varphi^{-1}P_a^{(\sigma)}D_\varphi.$$

Products of actions conjugate similarly. For any translation-homogeneous policy, $P_\pi$ is the same linear combination of $\{P_a\}$ and therefore also conjugate. Finally, resolvents preserve similarity:

$$M_\pi^{(\sigma')} = (I - \gamma P_\pi^{(\sigma')})^{-1} = (I - \gamma D_\varphi^{-1}P_\pi^{(\sigma)}D_\varphi)^{-1} = D_\varphi^{-1}(I - \gamma P_\pi^{(\sigma)})^{-1}D_\varphi = D_\varphi^{-1}M_\pi^{(\sigma)}D_\varphi.$$

$\square$

**D.2. Proof of Theorem 6.3**

*Proof of Theorem 6.3.* Assume there exist $g, h$ with $\beta_\sigma(g, h) \neq 1$, equivalently $U_g U_h = \beta_\sigma(g, h)\, U_h U_g$ by (23). Suppose for contradiction there exists a nonzero vector $v$ such that $U_g v = \lambda_g v$ and $U_h v = \lambda_h v$. Then

$$U_g U_h v = \lambda_g \lambda_h v \quad \text{and} \quad U_h U_g v = \lambda_h \lambda_g v = \lambda_g \lambda_h v,$$

but also $U_g U_h v = \beta_\sigma(g, h)\, U_h U_g v = \beta_\sigma(g, h)\lambda_g\lambda_h v$. Since $v \neq 0$ and $\lambda_g\lambda_h \neq 0$ (unitary operators have nonzero eigenvalues), this forces $\beta_\sigma(g, h) = 1$, a contradiction. Thus $U_g$ and $U_h$ share no eigenvector, and a fortiori the action family cannot admit a common eigenbasis as in Definition 2.1. $\square$

**D.3. Proof of Theorem 6.4**

*Proof of Theorem 6.4.* Let $U, V$ be the images of $U_{e_1}, U_{e_2}$ on some invariant block and assume $UV = \omega VU$ with $\omega = e^{2\pi i p/q}$ primitive. Choose an eigenvector $v$ of $V$ with eigenvalue $\lambda$ (exists over $\mathbb{C}$). By Lemma C.3, the vectors $\{U^t v\}_{t=0}^{q-1}$ are linearly independent, hence the representation dimension is at least $q$.

Moreover, $U$ maps the $\lambda$-eigenspace of $V$ to the $(\omega\lambda)$-eigenspace bijectively (since $U$ is invertible), so all eigenspaces in the orbit have equal dimension. Because the orbit has size $q$, the total dimension is a multiple of $q$, and in particular at least $q$. $\square$

**D.4. A cohomological impossibility corollary**

For completeness we record a corollary that makes explicit the "different holonomy class" obstruction.

**Corollary D.1** (Different holonomy classes are not gauge-conjugate). *If $[\sigma] \neq [\sigma']$ in $H^2(G, U(1))$, then there is no diagonal unitary $D$ such that $U_g' = D^{-1}U_g D$ for all $g \in G$.*

*Proof.* If such $D$ existed, then $\sigma$ and $\sigma'$ would differ by a coboundary (Lemma A.10), hence be cohomologous. Contradiction. $\square$

# E. Worked Examples and Explicit Constructions

This appendix makes the abstract machinery in Sections 4–6 concrete by working out the rational-flux torus model in full detail. The constructions below are the finite-torus analogue of magnetic translation symmetry and magnetic Bloch theory (Zak, 1964; Harper, 1955; Hofstadter, 1976). Throughout we emphasize explicit formulas that can be directly substituted into the twisted block formulas of Theorem 5.2 and Corollary 5.4.

**E.1. Rational flux on the discrete torus and a concrete cocycle**

Let $G = \mathbb{Z}_N^2$ with standard generators $e_1 = (1, 0)$ and $e_2 = (0, 1)$. Fix a *rational flux*

$$\Phi = \frac{p}{q}, \qquad \gcd(p, q) = 1, \qquad q \mid N,$$

and define $\omega \triangleq e^{2\pi i p/q}$ (a primitive $q$th root of unity). Write $N = qL$.

**Landau-gauge twisted translations.** Define operators $U_{e_1}, U_{e_2}$ on $\ell^2(G)$ by

$$(U_{e_1}f)(x_1, x_2) \triangleq f(x_1 + 1, x_2), \tag{30}$$

$$(U_{e_2}f)(x_1, x_2) \triangleq \exp\left(2\pi i \frac{p}{q} x_1\right) f(x_1, x_2 + 1), \tag{31}$$

with all coordinates understood modulo $N$. The phase in (30) is well-defined on $\mathbb{Z}_N$ since $x_1 \mapsto x_1 + N$ changes the exponent by $2\pi i(p/q)N = 2\pi i p L \in 2\pi i \mathbb{Z}$.

**Lemma E.1** (Constant plaquette holonomy). *The operators in* (30) *satisfy the Weyl commutation relation*

$$U_{e_1}U_{e_2} = \omega\, U_{e_2}U_{e_1}. \tag{32}$$

*Equivalently, the plaquette holonomy* $\mathcal{H}_{12}$ *defined in* (6) *equals* $\omega$ *at every plaquette.*

*Proof.* Compute

$$(U_{e_1}U_{e_2}f)(x_1, x_2) = \exp\left(2\pi i \frac{p}{q}(x_1 + 1)\right) f(x_1 + 1, x_2 + 1),$$

and

$$(U_{e_2}U_{e_1}f)(x_1, x_2) = \exp\left(2\pi i \frac{p}{q} x_1\right) f(x_1 + 1, x_2 + 1).$$

Taking the ratio yields the scalar factor $\exp(2\pi i p/q) = \omega$. $\qquad\qquad\square$

**A cocycle model.** The commutation (32) corresponds to a cocycle $\sigma_\Phi$ with commutator bicharacter $\beta_{\sigma_\Phi}(e_1, e_2) = \omega$ (Appendix B). A concrete representative is

$$\sigma_\Phi(g, h) \triangleq \omega^{-g_2 h_1}, \qquad g = (g_1, g_2),\ h = (h_1, h_2) \in \mathbb{Z}_N^2. \tag{33}$$

Then $\sigma_\Phi$ is a normalized 2-cocycle on $G$ and its commutator bicharacter satisfies

$$\beta_{\sigma_\Phi}(e_1, e_2) = \frac{\sigma_\Phi(e_1, e_2)}{\sigma_\Phi(e_2, e_1)} = \frac{1}{\omega^{-1}} = \omega.$$

By Proposition B.5, (33) represents the holonomy class determined by flux $\Phi = p/q$.

### E.2. Explicit projective irreducible representations and the block index set

We now write down all irreducible $\sigma_\Phi$-projective representations and show how they induce the $q \times q$ block structure of Corollary 5.5.

**Shift and clock matrices.** Let $X, Z \in \mathbb{C}^{q \times q}$ be defined on the standard basis $\{e_r\}_{r=0}^{q-1}$ by

$$Xe_r = e_{r+1\ (\mathrm{mod}\ q)}, \qquad Ze_r = \omega^{-r}e_r. \tag{34}$$

Then $X^q = Z^q = I$ and

$$XZ = \omega\, ZX. \tag{35}$$

**Definition E.2** (Canonical flux-$p/q$ irreps). Let $N = qL$. For each quasi-momentum $\kappa = (\kappa_1, \kappa_2) \in \mathbb{Z}_L^2$, define $\rho_\kappa : G \to U(q)$ by specifying generators

$$\rho_\kappa(e_1) \triangleq e^{2\pi i \kappa_1/N} X, \qquad \rho_\kappa(e_2) \triangleq e^{2\pi i \kappa_2/N} Z, \tag{36}$$

and extending to all $g = (g_1, g_2) \in \mathbb{Z}_N^2$ via

$$\rho_\kappa(g) \triangleq \rho_\kappa(e_1)^{g_1} \rho_\kappa(e_2)^{g_2}. \tag{37}$$

**Lemma E.3** ($\rho_\kappa$ is $\sigma_\Phi$-projective and respects torus periodicity). *Each $\rho_\kappa$ is a $\sigma_\Phi$-projective representation with cocycle (33). Moreover, since $N = qL$ and $X^q = Z^q = I$, one has $\rho_\kappa(Ne_1) = \rho_\kappa(Ne_2) = I$, so $\rho_\kappa$ is well-defined on $G = \mathbb{Z}_N^2$.*

*Proof.* Using (35) one checks that

$$\rho_\kappa(e_1)\rho_\kappa(e_2) = (e^{2\pi i(\kappa_1+\kappa_2)/N})XZ = \omega(e^{2\pi i(\kappa_1+\kappa_2)/N})ZX = \omega\,\rho_\kappa(e_2)\rho_\kappa(e_1),$$

so the commutator phase agrees with (32). More generally, from $Z^{g_2}X^{h_1} = \omega^{-g_2 h_1}X^{h_1}Z^{g_2}$ (iterating (35)) and (37) we get

$$\rho_\kappa(g)\rho_\kappa(h) = \omega^{-g_2 h_1}\rho_\kappa(g+h),$$

which is exactly the projective law (19) with cocycle $\sigma_\Phi(g,h) = \omega^{-g_2 h_1}$. Finally,

$$\rho_\kappa(Ne_1) = \left(e^{2\pi i\kappa_1/N}X\right)^N = e^{2\pi i\kappa_1}X^{qL} = I,$$

and similarly for $Ne_2$. □

**Lemma E.4** (Irreducibility). *Each $\rho_\kappa$ is irreducible.*

*Proof.* The matrices $X$ and $Z$ generate the full matrix algebra $\mathbb{C}^{q\times q}$: $Z$ has $q$ distinct eigenvalues, and $X$ cyclically permutes the eigenbasis, so any operator commuting with both must be diagonal in the $Z$-basis and invariant under cyclic permutation, hence a scalar multiple of $I$. Therefore the commutant of $\rho_\kappa(G)$ is $\mathbb{C}I$, and by Schur's lemma $\rho_\kappa$ is irreducible (Serre, 1977). □

**Proposition E.5** (Complete set of inequivalent irreps). *The family $\{\rho_\kappa : \kappa \in \mathbb{Z}_L^2\}$ is a complete set of inequivalent irreducible $\sigma_\Phi$-projective representations of $G$. In particular,*

$$|\widehat{G}_{\sigma_\Phi}| = L^2, \qquad d_{\rho_\kappa} = q, \qquad \sum_{\kappa\in\mathbb{Z}_L^2} d_{\rho_\kappa}^2 = L^2 q^2 = N^2 = |G|.$$

*Proof.* First, $\kappa$ is invariant under unitary equivalence because $\rho_\kappa(qe_1)$ and $\rho_\kappa(qe_2)$ are scalars:

$$\rho_\kappa(qe_1) = e^{2\pi i\kappa_1/L}I, \qquad \rho_\kappa(qe_2) = e^{2\pi i\kappa_2/L}I,$$

and scalars are preserved by conjugation. Thus $\rho_\kappa \simeq \rho_{\kappa'}$ implies $\kappa \equiv \kappa' \pmod{L}$.

Conversely, the count $\sum_\kappa d_{\rho_\kappa}^2 = |G|$ together with irreducibility (Lemma E.4) implies that these irreps exhaust $\widehat{G}_{\sigma_\Phi}$ (by the standard dimension-sum identity for semisimple finite-dimensional $*$-algebras; see (Karpilovsky, 1985; Terras, 1999)). □

Proposition E.5 is the fully explicit version of Corollary 5.5: rational flux $\Phi = p/q$ forces $q$-dimensional spectral blocks, and the blocks are indexed by a reduced Brillouin zone $\kappa \in \mathbb{Z}_L^2$.

### E.3. Magnetic Bloch waves as explicit holonomy grid codes

The twisted Fourier basis functions are matrix coefficients of $\rho_\kappa$ (Section 5.4). Here we write them in closed form and record their quasi-periodicity properties, which justify the term *magnetic Bloch waves*.

**Definition E.6** (Magnetic Bloch (matrix-coefficient) basis). For $\kappa \in \mathbb{Z}_L^2$ and indices $u, v \in \{0, \ldots, q-1\}$, define

$$\phi_{\kappa,u,v}(g) \triangleq \sqrt{\frac{q}{|G|}}\,\left[\rho_\kappa(g)\right]_{u,v}, \qquad g \in \mathbb{Z}_N^2, \tag{38}$$

where $\rho_\kappa$ is given by (37).

**Lemma E.7** (Closed form). *Let $g = (g_1, g_2) \in \mathbb{Z}_N^2$. Then*

$$\phi_{\kappa,u,v}(g_1, g_2) = \sqrt{\frac{q}{N^2}}\,\exp\!\left(\frac{2\pi i}{N}(\kappa_1 g_1 + \kappa_2 g_2)\right)\omega^{-vg_2}\,\mathbf{1}\{u \equiv v + g_1 \pmod{q}\}. \tag{39}$$

*Proof.* From (37) and (34), we have

$$\rho_\kappa(g_1, g_2) = \exp\left(\frac{2\pi i}{N}(\kappa_1 g_1 + \kappa_2 g_2)\right) X^{g_1} Z^{g_2}.$$

Since $Z^{g_2} e_v = \omega^{-vg_2} e_v$ and $X^{g_1} e_v = e_{v+g_1}$, the matrix entry equals $\left[X^{g_1} Z^{g_2}\right]_{u,v} = \omega^{-vg_2} \mathbf{1}\{u \equiv v + g_1 \pmod{q}\}$, yielding (39). $\square$

**Quasi-periodicity.** Equation (39) implies a magnetic Bloch-type quasi-periodicity over the enlarged unit cell:

**Lemma E.8** (Magnetic Bloch quasi-periodicity). *For all $g_2 \in \mathbb{Z}_N$ and $g_1 \in \mathbb{Z}_N$,*

$$\phi_{\kappa,u,v}(g_1 + q, g_2) = \exp\left(\frac{2\pi i}{L}\kappa_1\right) \phi_{\kappa,u,v}(g_1, g_2), \tag{40}$$

$$\phi_{\kappa,u,v}(g_1, g_2 + q) = \exp\left(\frac{2\pi i}{L}\kappa_2\right) \phi_{\kappa,u,v}(g_1, g_2). \tag{41}$$

*Proof.* Use $N = qL$ in the plane-wave prefactor of (39) and the fact that $\omega^{-v(g_2+q)} = \omega^{-vg_2}$ because $\omega^q = 1$. $\square$

Thus the relevant "crystal momentum" lives on the reduced torus $\mathbb{Z}_L^2$ and the internal indices $(u, v)$ account for the $q$-fold magnetic enlargement imposed by flux $\Phi = p/q$.

### E.4. Explicit block multipliers for common action kernels

Let $k : G \to \mathbb{C}$ be a kernel and $L_k$ the corresponding left convolution operator $f \mapsto k *_{\sigma_\Phi} f$ (Definition A.2). In the flux-$p/q$ model, Proposition E.5 lets us index blocks by $\kappa \in \mathbb{Z}_L^2$ with block size $q$.

**Proposition E.9** (Explicit block multiplier formula). *For every kernel $k : G \to \mathbb{C}$ and every $\kappa \in \mathbb{Z}_L^2$, the twisted Fourier block multiplier equals*

$$\widehat{k}(\kappa) = \sum_{g \in G} k(g)\, \rho_\kappa(g) = \sum_{g_1, g_2 = 0}^{N-1} k(g_1, g_2)\, \exp\left(\frac{2\pi i}{N}(\kappa_1 g_1 + \kappa_2 g_2)\right) X^{g_1} Z^{g_2} \in \mathbb{C}^{q \times q}. \tag{42}$$

*Moreover, for any signal $f$,*

$$(\mathcal{F}_{\sigma_\Phi}(L_k f))(\kappa) = \widehat{k}(\kappa)\, (\mathcal{F}_{\sigma_\Phi} f)(\kappa),$$

*which is the specialization of Theorem 5.2.*

**Deterministic directed actions.** Let $k_E = \delta_{e_1}$ (one-step East) and $k_N = \delta_{e_2}$ (one-step North). Then

$$\widehat{k_E}(\kappa) = \rho_\kappa(e_1) = e^{2\pi i \kappa_1 / N} X, \qquad \widehat{k_N}(\kappa) = \rho_\kappa(e_2) = e^{2\pi i \kappa_2 / N} Z. \tag{43}$$

These matrices do not commute when $\omega \neq 1$, recovering the impossibility of scalar basis reuse (Theorem 6.3) and motivating block reuse instead.

**Isotropic diffusion (nearest-neighbor random walk).** Let $\varepsilon \in (0, 1/4)$ and define

$$k_{\text{diff}}(0) = 1 - 4\varepsilon, \quad k_{\text{diff}}(\pm e_1) = \varepsilon, \quad k_{\text{diff}}(\pm e_2) = \varepsilon, \quad k_{\text{diff}}(g) = 0 \text{ otherwise.}$$

Then (42) yields

$$\widehat{k_{\text{diff}}}(\kappa) = (1 - 4\varepsilon)I + \varepsilon\left(e^{2\pi i \kappa_1 / N} X + e^{-2\pi i \kappa_1 / N} X^{-1} + e^{2\pi i \kappa_2 / N} Z + e^{-2\pi i \kappa_2 / N} Z^{-1}\right). \tag{44}$$

For fixed $\kappa$, (44) is exactly the $q \times q$ *Harper matrix* associated to flux $p/q$ (Harper, 1955); its spectral dependence on $\Phi$ underlies the Hofstadter butterfly in the corresponding tight-binding Hamiltonian (Hofstadter, 1976).

**Directed drift + diffusion.** If one adds directional bias (e.g., $\varepsilon_+ \neq \varepsilon_-$ on $\pm e_1$), the multiplier becomes the same noncommutative polynomial but with asymmetric coefficients. This is a useful template for controlled transitions where each action corresponds to a distinct drift vector plus shared diffusion.

### E.5. Successor blocks in the rational-flux model

Under a translation-homogeneous policy $\pi$, the induced kernel is $k_\pi = \sum_a \pi(a) k_a$ and the SR blocks are

$$\widehat{M_\pi}(\kappa) = \left(I - \gamma \widehat{k_\pi}(\kappa)\right)^{-1} \in \mathbb{C}^{q \times q}.$$

In the diffusion example (44), each SR block is the inverse of a shifted Harper matrix. This makes the curvature/flux tradeoff completely explicit: exact prediction and SR computation reduce to inverting $q \times q$ matrices over $\kappa \in \mathbb{Z}_L^2$, and Theorem 6.4 shows $q$ is information-theoretically unavoidable when the holonomy has order $q$.

## F. Robustness and Perturbation Theory

Sections 4–6 analyze an *ideal* holonomy MDP in which each action operator lies exactly in the twisted translation algebra $\mathcal{A}_\sigma$ and therefore admits a reused block spectral decomposition (Theorem 5.2). Realistic environments introduce deviations: walls remove transitions locally, boundaries break translation-homogeneity, and policies may vary across space. This appendix quantifies how the key objects—multi-step prediction and the successor resolvent—change under such deviations, and it gives constructive low-rank update formulas that preserve most of the computational benefit of the twisted decomposition.

Throughout, we work with any submultiplicative operator norm $\|\cdot\|$ (e.g., spectral norm). All bounds below are deterministic and do not rely on stochasticity or reversibility.

### F.1. Perturbations of prediction operators and resolvents

Let $P_0$ denote a baseline operator (e.g., a translation-homogeneous policy operator) and let $E$ be a defect:

$$P = P_0 + E. \tag{45}$$

We assume $P_0 \in \mathcal{A}_\sigma$ so that $P_0$ is block diagonalizable by $\mathcal{F}_\sigma$ (Theorem 5.2), while $E$ may be arbitrary.

**Lemma F.1** (Neumann series and basic bounds). *If $\gamma \|P\| < 1$, then the successor resolvent exists and*

$$M(P) \triangleq (I - \gamma P)^{-1} = \sum_{t=0}^{\infty} \gamma^t P^t, \qquad \|M(P)\| \leq \frac{1}{1 - \gamma \|P\|}.$$

*Proof.* Submultiplicativity gives $\sum_{t \geq 0} \|\gamma^t P^t\| \leq \sum_{t \geq 0} (\gamma \|P\|)^t < \infty$. The series is the inverse by direct multiplication, and the norm bound follows. $\qquad \square$

**Lemma F.2** (Resolvent identity). *If $I - \gamma P_0$ and $I - \gamma(P_0 + E)$ are invertible, then*

$$M(P_0 + E) - M(P_0) = M(P_0 + E)(\gamma E) M(P_0). \tag{46}$$

*Proof.* Use $(A + B)^{-1} - A^{-1} = (A + B)^{-1}(-B) A^{-1}$ with $A = I - \gamma P_0$ and $B = -\gamma E$. $\qquad \square$

**Proposition F.3** (Lipschitz bound for the successor operator). *Assume $\gamma \|P_0\| < 1$ and $\gamma \|P_0 + E\| < 1$. Then*

$$\|M(P_0 + E) - M(P_0)\| \leq \frac{\gamma \|E\|}{(1 - \gamma \|P_0 + E\|)(1 - \gamma \|P_0\|)}. \tag{47}$$

*In particular, if $\|P_0\| \leq 1$ and $\|P_0 + E\| \leq 1 + \|E\|$, then*

$$\|M(P_0 + E) - M(P_0)\| \leq \frac{\gamma \|E\|}{(1 - \gamma(1 + \|E\|))(1 - \gamma)}.$$

*Proof.* Combine Lemma F.2 and Lemma F.1:

$$\|M(P_0 + E) - M(P_0)\| \leq \|M(P_0 + E)\| \gamma \|E\| \|M(P_0)\| \leq \frac{1}{1 - \gamma \|P_0 + E\|} \gamma \|E\| \frac{1}{1 - \gamma \|P_0\|}.$$

$\qquad \square$

**Corollary F.4** (Perturbation of $t$-step prediction). *For any integer $t \geq 1$,*

$$\|P^t - P_0^t\| \ \leq \ \sum_{j=0}^{t-1} \|P\|^{t-1-j} \|E\| \|P_0\|^j \ \leq \ t\|E\| \max\{\|P\|, \|P_0\|\}^{t-1}.$$

*Proof.* Use the telescoping identity $P^t - P_0^t = \sum_{j=0}^{t-1} P^{t-1-j}(P - P_0)P_0^j$ and submultiplicativity. $\square$

**Interpretation.** Proposition F.3 and Corollary F.4 formalize a simple message: if an environment is "mostly" a holonomy MDP with a small defect operator $E$, then the twisted spectral prediction computed from $P_0 \in \mathcal{A}_\sigma$ remains accurate, and the SR error scales at most linearly in $\|E\|$ (for fixed $\gamma$ away from 1).

## F.2. Low-rank and row-local defects via Woodbury

Obstacles and boundaries are typically *localized*: they modify transition rules only on a small set of states. This locality yields low-rank structure that can be exploited to compute the exact SR with a small correction on top of the block-diagonalizable baseline.

**Proposition F.5** (Woodbury update for successor resolvents). *Let $P = P_0 + E$ with $P_0 \in \mathbb{R}^{n \times n}$ and suppose $E$ admits a rank-$r$ factorization $E = UV^\top$ with $U, V \in \mathbb{R}^{n \times r}$. Assume $I - \gamma P_0$ is invertible and define $M_0 = (I - \gamma P_0)^{-1}$. If $I_r - \gamma V^\top M_0 U$ is invertible, then*

$$M(P) \ = \ M_0 \ + \ \gamma M_0 U (I_r - \gamma V^\top M_0 U)^{-1} V^\top M_0. \tag{48}$$

*Proof.* Apply the Woodbury identity to $I - \gamma(P_0 + UV^\top) = (I - \gamma P_0) - \gamma UV^\top$; see, e.g., (Golub & Loan, 2013). $\square$

**Lemma F.6** (Row-local defects have low rank). *Let $E \in \mathbb{R}^{n \times n}$ have support on at most $m$ rows (i.e., $E_{s,\cdot} = 0$ for all but $m$ states $s$). Then $\mathrm{rank}(E) \leq m$, and $E$ can be written as $E = UV^\top$ with $U \in \mathbb{R}^{n \times m}$ containing the corresponding standard basis vectors.*

*Proof.* Write $E = \sum_{j=1}^m e_{s_j} \Delta_j^\top$ where $e_{s_j}$ is the $s_j$th standard basis vector and $\Delta_j^\top$ is the modified row; this is a sum of $m$ rank-1 matrices. $\square$

**Obstacle model as row-local defect.** A common way to incorporate walls is to start from a translation-homogeneous baseline kernel and then *renormalize* transition probabilities in states adjacent to blocked edges. This changes only the rows corresponding to boundary-adjacent states; hence Lemma F.6 applies with $m$ equal to the number of affected states. Proposition F.5 then yields an exact SR correction requiring only an $m \times m$ matrix inversion, assuming we can apply $M_0$ efficiently (which we can using the twisted block decomposition of $P_0$).

## F.3. Block-structured approximations in the twisted Fourier domain

Let $\mathcal{F}_\sigma$ be the unitary transform from Theorem 5.2, so that $\mathcal{F}_\sigma P_0 \mathcal{F}_\sigma^{-1}$ is block diagonal. Define the defect in this basis:

$$\widetilde{E} \ \triangleq \ \mathcal{F}_\sigma E \mathcal{F}_\sigma^{-1}.$$

Even if $E$ is sparse in the original state basis, $\widetilde{E}$ generally introduces couplings *between* blocks. A natural approximation is to retain only the within-block part:

$$\widetilde{E}_{\mathrm{diag}} \ \triangleq \ \mathrm{blkdiag}\big(\widetilde{E}_{\rho\rho}\big)_{\rho \in \widehat{G}_\sigma}, \qquad E_{\mathrm{diag}} \ \triangleq \ \mathcal{F}_\sigma^{-1} \widetilde{E}_{\mathrm{diag}} \mathcal{F}_\sigma,$$

and define the "block-preserving" approximation $P_{\mathrm{diag}} = P_0 + E_{\mathrm{diag}}$.

**Proposition F.7** (Error from ignoring cross-block couplings). *Assume $\gamma\|P\| < 1$ and $\gamma\|P_{\mathrm{diag}}\| < 1$. Then*

$$\|M(P) - M(P_{\mathrm{diag}})\| \ \leq \ \frac{\gamma\|E - E_{\mathrm{diag}}\|}{(1 - \gamma\|P\|)(1 - \gamma\|P_{\mathrm{diag}}\|)}.$$

*Proof.* Apply Proposition F.3 with $P_0 \leftarrow P_{\mathrm{diag}}$ and defect $E \leftarrow P - P_{\mathrm{diag}} = E - E_{\mathrm{diag}}$. $\square$

**Interpretation.** Proposition F.7 isolates the *cross-block* component $E - E_{\text{diag}}$ as the true source of error when one insists on preserving block reuse. This formalizes a practical principle: if defects are localized and do not strongly mix twisted Fourier blocks, then a block-preserving approximation can remain accurate.

### F.4. Truncated-horizon SR approximation

Finally, if one prefers a finite-horizon approximation to the SR,

$$M^{(T)}(P) \triangleq \sum_{t=0}^{T} \gamma^t P^t,$$

the tail error is controlled uniformly.

**Lemma F.8** (SR truncation error). *If $\gamma\|P\| < 1$, then*

$$\|M(P) - M^{(T)}(P)\| \leq \sum_{t=T+1}^{\infty} \gamma^t \|P\|^t = \frac{(\gamma\|P\|)^{T+1}}{1 - \gamma\|P\|}.$$

*Proof.* Immediate from the Neumann series and geometric summation. $\qquad\square$

## G. Explicit Algorithms

Here we provide explicit procedures for applying action operators, multi-step prediction, and successor resolvents using the twisted block decomposition. We present (i) a generic algorithm given access to irreducible $\sigma$-projective representations and (ii) a specialized fast pipeline for the rational-flux torus where irreps are explicit (Appendix E). These algorithms are included to make the theory constructive; no empirical evaluation is required for their correctness.

### G.1. Generic twisted Fourier pipeline

Fix a cocycle $\sigma$ on $G$ and a complete set of irreducible $\sigma$-projective representations $\widehat{G}_\sigma = \{\rho\}$ with dimensions $d_\rho$. Use the main-text convention

$$(\mathcal{F}_\sigma f)(\rho) \triangleq \sqrt{\frac{d_\rho}{|G|}} \sum_{x \in G} f(x)\,\rho(x)^* \in \mathbb{C}^{d_\rho \times d_\rho}. \tag{49}$$

The inverse transform is

$$f(x) = \sum_{\rho \in \widehat{G}_\sigma} \sqrt{\frac{d_\rho}{|G|}} \operatorname{Tr}\big(\rho(x)\,(\mathcal{F}_\sigma f)(\rho)\big), \tag{50}$$

which follows from projective Schur orthogonality (Theorem A.11).

Each action operator is specified by a kernel $k_a : G \to \mathbb{R}_{\geq 0}$ and acts by left twisted convolution $P_a f = k_a *_\sigma f$. Its block multiplier is

$$\widehat{k_a}(\rho) \triangleq \sum_{g \in G} k_a(g)\,\rho(g) \in \mathbb{C}^{d_\rho \times d_\rho}.$$

Then Theorem 5.2 says $(\mathcal{F}_\sigma(P_a f))(\rho) = \widehat{k_a}(\rho)(\mathcal{F}_\sigma f)(\rho)$.

**Proposition G.1** (Naive complexity accounting). *Assume explicit access to matrices $\rho(x)$ for all $x \in G$ and $\rho \in \widehat{G}_\sigma$.*

- *Computing $\mathcal{F}_\sigma f$ naively from (49) costs $O\big(|G| \sum_\rho d_\rho^2\big)$ arithmetic operations.*

- *Applying block multipliers costs $O\big(\sum_\rho d_\rho^3\big)$.*

- *Precomputing all action multipliers $\widehat{k_a}(\rho)$ costs $O\big(|\mathcal{A}|\,|G| \sum_\rho d_\rho^2\big)$.*

*Since $\sum_\rho d_\rho^2 = |G|$, the naive transform cost is $O(|G|^2)$.*

---

**Algorithm 1** Apply a single action operator $P_a$ via twisted blocks

---

**Require:** Signal $f : G \to \mathbb{C}$, cocycle $\sigma$, irreps $\{\rho\}$, precomputed multipliers $\{\widehat{k_a}(\rho)\}_\rho$
**Ensure:** $y = P_a f$
1: Compute block coefficients $\widehat{f}(\rho) \leftarrow (\mathcal{F}_\sigma f)(\rho)$ for all $\rho$ using (49)
2: For each $\rho$, set $\widehat{y}(\rho) \leftarrow \widehat{k_a}(\rho)\,\widehat{f}(\rho)$
3: Reconstruct $y \leftarrow \mathcal{F}_\sigma^{-1}\widehat{y}$ using (50)
   **return** $y$

---

---

**Algorithm 2** Apply the successor operator $M_\pi = (I - \gamma P_\pi)^{-1}$ under a translation-homogeneous policy

---

**Require:** Signal $f : G \to \mathbb{C}$, discount $\gamma \in (0,1)$, policy weights $\pi(a)$, precomputed $\widehat{k_a}(\rho)$
**Ensure:** $y = M_\pi f$
1: For each $\rho$, form $\widehat{k_\pi}(\rho) \leftarrow \sum_a \pi(a)\widehat{k_a}(\rho)$
2: Compute $\widehat{f}(\rho) \leftarrow (\mathcal{F}_\sigma f)(\rho)$ for all $\rho$
3: For each $\rho$, solve the $d_\rho \times d_\rho$ linear system

$$\widehat{y}(\rho) \leftarrow (I - \gamma\widehat{k_\pi}(\rho))^{-1}\widehat{f}(\rho)$$

4: Reconstruct $y \leftarrow \mathcal{F}_\sigma^{-1}\widehat{y}$ using (50)
   **return** $y$

---

*Proof.* Direct from matrix accumulation costs per term and the dimension identity $\sum_\rho d_\rho^2 = |G|$ (Theorem A.11). $\square$

Proposition G.1 clarifies why structure matters: the twisted decomposition dramatically reduces the *per-block* operations (only small $d_\rho \times d_\rho$ inversions), but computing the transform itself naively is quadratic in $|G|$. For structured groups (e.g., tori with rational flux), one can exploit additional algebra to compute the transform faster, analogous to FFT acceleration in the untwisted case (Cooley & Tukey, 1965).

### G.2. Specialized fast pipeline for rational flux on $\mathbb{Z}_N^2$

In Appendix E we constructed an explicit complete set of irreps $\{\rho_\kappa\}_{\kappa \in \mathbb{Z}_L^2}$ for flux $\Phi = p/q$ with $N = qL$, each of dimension $q$. This explicit parameterization yields a practical "twisted FFT" viewpoint: blocks are indexed by $\kappa$ (a reduced Brillouin zone), and each block carries an internal $q$-dimensional fiber.

We state a concrete algorithmic reduction without overcommitting to a particular implementation detail.

**Proposition G.2** (Flux-$p/q$ block structure enables FFT-based acceleration). *Let $G = \mathbb{Z}_N^2$ with $N = qL$ and flux $\Phi = p/q$, and let $\widehat{G}_{\sigma_\Phi} = \{\rho_\kappa : \kappa \in \mathbb{Z}_L^2\}$ as in Proposition E.5. Then for any signal $f : G \to \mathbb{C}$, the map $f \mapsto \{\widehat{f}(\kappa)\}_{\kappa \in \mathbb{Z}_L^2}$ with*

$$\widehat{f}(\kappa) \;=\; \sum_{x \in G} f(x)\,\rho_\kappa(x)^* \in \mathbb{C}^{q \times q}$$

*can be computed by combining: (i) standard FFTs over the $L \times L$ reduced momentum grid, (ii) q-dimensional linear mixing in the internal indices, and (iii) pointwise phase factors determined by the Landau-gauge form of $\rho_\kappa$.*

**Remark.** Proposition G.2 is the flux analogue of the standard statement that convolution on $\mathbb{Z}_N^2$ is diagonalized by a 2D FFT. Here, diagonalization becomes $q \times q$ block diagonalization; the FFT still handles the global momentum indices $\kappa$, while the internal $q$ dimension accounts for holonomy (Theorem 6.4).

### G.3. Low-rank corrected SR algorithm

When the environment differs from a holonomy baseline by a row-local defect (Lemma F.6), one can compute the exact SR by combining Algorithm 2 with Woodbury.

---

**Algorithm 3** Exact SR under a low-rank defect $P = P_0 + UV^\top$

---

**Require:** Baseline block-diagonalizable $P_0 \in \mathcal{A}_\sigma$, defect factors $U, V \in \mathbb{R}^{n \times r}$, discount $\gamma$, input $f$
**Ensure:** $y = (I - \gamma P)^{-1} f$
  1: Compute $M_0 f$ via Algorithm 2 (with $P_0$ in place of $P_\pi$)
  2: Compute $A \leftarrow V^\top (M_0 U) \in \mathbb{R}^{r \times r}$ by applying $M_0$ to each column of $U$
  3: Solve the $r \times r$ system $(I_r - \gamma A)z = V^\top (M_0 f)$
  4: Output $y \leftarrow M_0 f + \gamma(M_0 U)z$ (from (48))
     **return** $y$

---

This procedure is exact whenever the defect is exactly low rank; in obstacle models, row locality often makes $r$ equal to the number of affected states, which can be far smaller than $|G|$.

## H. Stronger Impossibility and Lower Bounds

Section 6 established that nontrivial holonomy obstructs *scalar* basis reuse and forces block structure; in the constant-flux case, the minimal block size is at least $q$ (Theorem 6.4). This appendix strengthens these claims in three directions: (i) the dimension must be a multiple of $q$ (not merely $\geq q$), (ii) an *approximate* Weyl relation already forces large dimension when the error is small, and (iii) a Weyl pair cannot be uniformly approximated by any commuting family without incurring a constant error depending on holonomy.

### H.1. Exact Weyl relations force dimension to be a multiple of $q$

**Theorem H.1** (Multiplicity of the flux denominator). *Let $U, V$ be unitary operators on a finite-dimensional complex vector space $\mathcal{H}$ satisfying*

$$UV = \omega VU, \tag{51}$$

*where $\omega$ is a primitive qth root of unity. Then $\dim(\mathcal{H})$ is a multiple of q.*

*Proof.* Because $V$ is unitary, $\mathcal{H}$ decomposes as a direct sum of eigenspaces of $V$:

$$\mathcal{H} = \bigoplus_{\lambda \in \mathrm{spec}(V)} \mathcal{E}_\lambda, \qquad \mathcal{E}_\lambda \triangleq \{x : Vx = \lambda x\}.$$

From (51), we obtain $V(Ux) = \omega^{-1} U(Vx)$, hence $U$ maps $\mathcal{E}_\lambda$ bijectively onto $\mathcal{E}_{\omega^{-1}\lambda}$. Therefore, for each eigenvalue $\lambda$, the orbit

$$\lambda, \ \omega^{-1}\lambda, \ \omega^{-2}\lambda, \ \ldots, \ \omega^{-(q-1)}\lambda$$

consists of $q$ distinct eigenvalues (since $\omega$ is primitive), and all eigenspaces along the orbit have the same dimension. Summing dimensions along each orbit yields a multiple of $q$, and the full dimension is a sum of orbit-sums, hence is a multiple of $q$. $\qquad\square$

**Consequence for block decompositions.** If a representation of the constant-flux algebra decomposes into invariant blocks, each block carries a representation of the generators and must therefore have dimension a multiple of $q$. This strengthens the "minimal block size" intuition: holonomy not only forces $q$ as a minimum, it forces $q$ as the atomic unit of invariant subspaces.

### H.2. Approximate Weyl relations imply a quantitative dimension lower bound

We next show that *approximate* holonomy already enforces large dimension if the approximation error is small compared to the separation between the $q$ distinct phases.

**Lemma H.2** (Residual implies nearby spectrum for normal matrices). *Let $A$ be normal (in particular, unitary) on $\mathcal{H}$ and let $w \neq 0$. If $\|(A - \mu I)w\| \leq \delta\|w\|$, then $\mathrm{dist}(\mu, \mathrm{spec}(A)) \leq \delta$.*

*Proof.* If $\mathrm{dist}(\mu, \mathrm{spec}(A)) > \delta$, then $A - \mu I$ is invertible and $\|(A - \mu I)^{-1}\| = 1/\mathrm{dist}(\mu, \mathrm{spec}(A)) < 1/\delta$ for normal $A$. Hence

$$\|w\| = \|(A - \mu I)^{-1}(A - \mu I)w\| \leq \|(A - \mu I)^{-1}\| \cdot \|(A - \mu I)w\| < \frac{1}{\delta} \cdot \delta \|w\| = \|w\|,$$

a contradiction. $\square$

**Lemma H.3** (Orbit residual bound under approximate Weyl relations). *Let $U, V$ be unitary and suppose*

$$\|UV - \omega VU\| \leq \varepsilon. \tag{52}$$

*Let $v$ be any eigenvector of $V$ with $Vv = \lambda v$ and define $v_t \triangleq U^t v$. Then for each $t \geq 0$,*

$$\|Vv_t - \omega^{-t} \lambda v_t\| \leq t \varepsilon \|v\|. \tag{53}$$

*Proof.* From (52) we have $VU = \omega^{-1} UV + E$ where $E \triangleq -\omega^{-1}(UV - \omega VU)$ and $\|E\| \leq \varepsilon$. A simple induction using $VU^{t+1} = (VU^t)U$ yields

$$VU^t = \omega^{-t} U^t V + \sum_{j=0}^{t-1} \omega^{-(j+1)} U^j E U^{t-1-j}.$$

Applying this identity to $v$ and using $\|U^j E U^{t-1-j} v\| \leq \|E\| \|v\|$ gives (53). $\square$

**Theorem H.4** (Approximate holonomy forces large dimension). *Let $U, V$ be unitary on a $d$-dimensional space and let $\omega$ be a primitive $q$th root of unity with $q \geq 2$. Suppose $\|UV - \omega VU\| \leq \varepsilon$. If*

$$(q - 1)\varepsilon < \sin\left(\frac{\pi}{q}\right), \tag{54}$$

*then $d \geq q$.*

*Proof.* Pick any eigenpair $Vv = \lambda v$ with $\|v\| = 1$ and define $v_t = U^t v$ for $t = 0, \dots, q - 1$. By Lemma H.3, each $v_t$ is a $(t\varepsilon)$-approximate eigenvector of $V$ with target eigenvalue $\mu_t \triangleq \omega^{-t} \lambda$:

$$\|(V - \mu_t I)v_t\| \leq t\varepsilon.$$

By Lemma H.2, $\mathrm{spec}(V)$ contains an eigenvalue within distance $t\varepsilon$ of $\mu_t$. In particular, it contains an eigenvalue within distance $(q - 1)\varepsilon$ of each $\mu_t$.

The $q$ points $\{\mu_t\}_{t=0}^{q-1}$ lie on the unit circle and are equally spaced by angle $2\pi/q$, hence their minimum Euclidean separation is

$$\min_{t \neq t'} |\mu_t - \mu_{t'}| = |1 - \omega| = 2\sin\left(\frac{\pi}{q}\right).$$

Under condition (54), the closed disks of radius $(q - 1)\varepsilon$ around the $\mu_t$ are pairwise disjoint. Therefore, $\mathrm{spec}(V)$ must contain at least $q$ *distinct* eigenvalues, which is only possible if $d \geq q$. $\square$

**Discussion.** Theorem H.4 strengthens Theorem 6.4 by giving an explicit robustness margin: even if holonomy is only approximately satisfied (in operator norm), small error still forces the same representational dimension.

### H.3. Distance to commuting families

A natural "failed alternative" to holonomy-aware blocks is to approximate the holonomy dynamics by an untwisted (commutative) model, i.e., by commuting operators. The next bound shows that this requires a non-vanishing error depending on the holonomy phase.

**Theorem H.5** (Weyl pairs are bounded away from commuting pairs). *Let $U, V$ be unitary and satisfy $UV = \omega VU$ with $\omega \neq 1$. Let $A, B$ be commuting matrices with $\|A\| \leq 1$ and $\|B\| \leq 1$, and define $\Delta_U \triangleq U - A$ and $\Delta_V \triangleq V - B$. Then*

$$\|\Delta_U\| \vee \|\Delta_V\| \geq -1 + \sqrt{1 + \frac{|1 - \omega|}{2}}. \tag{55}$$

*In particular, for $|1 - \omega| \leq 2$ one has the simpler bound*

$$\|\Delta_U\| \vee \|\Delta_V\| \gtrsim \frac{|1 - \omega|}{4},$$

*up to higher-order terms in $|1 - \omega|$.*

*Proof.* Write $U = A + \Delta_U$ and $V = B + \Delta_V$ with $AB = BA$. Expand

$$UV - VU = (A + \Delta_U)(B + \Delta_V) - (B + \Delta_V)(A + \Delta_U)$$

and cancel $AB - BA = 0$ to obtain

$$UV - VU = (A\Delta_V - \Delta_V A) + (\Delta_U B - B\Delta_U) + (\Delta_U \Delta_V - \Delta_V \Delta_U).$$

Taking norms and using $\|A\|, \|B\| \leq 1$ gives

$$\|UV - VU\| \leq 2\|\Delta_V\| + 2\|\Delta_U\| + 2\|\Delta_U\| \, \|\Delta_V\|.$$

Let $\delta \triangleq \|\Delta_U\| \vee \|\Delta_V\|$. Then $\|UV - VU\| \leq 4\delta + 2\delta^2$.

Under the Weyl relation $UV = \omega VU$, we have $UV - VU = (\omega - 1)VU$, hence $\|UV - VU\| = |1 - \omega|$ because $VU$ is unitary. Therefore $|1 - \omega| \leq 4\delta + 2\delta^2$, which rearranges to $\delta \geq -1 + \sqrt{1 + |1 - \omega|/2}$. $\qquad \square$

**Implication.** Theorem H.5 provides a clean "no-free-lunch" statement: nontrivial holonomy cannot be captured by any commuting (i.e., scalar Fourier) approximation without incurring a constant operator-norm error. This formalizes why the block upgrade is not merely convenient but necessary.

## I. Additional Worked Examples Beyond Constant Flux

Appendix E worked out the constant-flux (2D rational $\Phi = p/q$) model in detail. This appendix provides additional explicit constructions that (i) illustrate gauge equivalence at the operator level, (ii) show how higher-dimensional holonomy decomposes into independent flux planes, and (iii) connect the block sizes in Theorem 6.4 to a general "radical size" formula.

### I.1. Gauge-equivalent vector potentials for the same flux

On $\mathbb{Z}_N^2$, the constant-flux holonomy class can be realized by many different "gauges" (phase conventions). Here we show an explicit diagonal gauge that maps Landau gauge (Appendix E.1) to a symmetric gauge.

Fix $\Phi = p/q$ with $N = qL$ and define $\omega = e^{2\pi i p/q}$. Let $(U_{e_1}, U_{e_2})$ be Landau-gauge translations as in (30).

Define alternative generators $(V_{e_1}, V_{e_2})$ by

$$(V_{e_1} f)(x_1, x_2) \triangleq \exp\left(-\pi i \frac{p}{q} x_2\right) f(x_1 + 1, x_2), \tag{56}$$

$$(V_{e_2} f)(x_1, x_2) \triangleq \exp\left(\pi i \frac{p}{q} x_1\right) f(x_1, x_2 + 1). \tag{57}$$

**Lemma I.1** (Same holonomy, different gauge). *The symmetric-gauge generators (56) satisfy $V_{e_1} V_{e_2} = \omega V_{e_2} V_{e_1}$, hence define the same constant flux. Moreover, there exists a diagonal unitary $D$ such that $V_{e_i} = D^{-1} U_{e_i} D$, $i = 1, 2$.*

*Proof.* The commutator calculation is immediate by comparing phase differences, exactly as in Lemma E.1. For gauge equivalence, define

$$(Df)(x_1, x_2) \triangleq \exp\left(\pi i \frac{p}{q} x_1 x_2\right) f(x_1, x_2).$$

A direct substitution shows $D^{-1} U_{e_1} D$ and $D^{-1} U_{e_2} D$ match (56). $\qquad \square$

Lemma I.1 is the explicit instance of the general gauge-transfer theorem (Theorem 6.2): the physics/geometry is captured by holonomy, not by a particular choice of edge phases.

## I.2. A $3$D example: flux in one plane and a commuting direction

Let $G = \mathbb{Z}_N^3$ with generators $e_1, e_2, e_3$. Impose constant flux $\Phi = p/q$ only in the $(1,2)$-plane:

$$U_{e_1}U_{e_2} = \omega U_{e_2}U_{e_1}, \qquad U_{e_1}U_{e_3} = U_{e_3}U_{e_1}, \qquad U_{e_2}U_{e_3} = U_{e_3}U_{e_2}.$$

One concrete realization is to take Landau-gauge phases depending on $x_1$ exactly as in 2D, and let $U_{e_3}$ be a pure shift in $x_3$.

**Proposition I.2** (Irrep factorization for 3D partial flux). *Assume $N = qL$ and flux $\Phi = p/q$ in the $(1,2)$ plane. Then every irreducible projective representation factors as*

$$\rho_{\kappa,\eta}(g_1, g_2, g_3) \simeq \rho_\kappa^{(12)}(g_1, g_2) \otimes \chi_\eta(g_3),$$

*where $\rho_\kappa^{(12)}$ is a $q$-dimensional 2D flux irrep indexed by $\kappa \in \mathbb{Z}_L^2$ (Appendix E.2) and $\chi_\eta(g_3) = \exp(2\pi i \eta g_3/N)$ is an ordinary character indexed by $\eta \in \mathbb{Z}_N$. Consequently, all irreps have dimension $q$, and the block index set is $\mathbb{Z}_L^2 \times \mathbb{Z}_N$.*

*Proof.* Because the $e_3$-direction commutes with $e_1, e_2$, it generates a commuting subgroup whose action can be simultaneously diagonalized within each irreducible block; hence it contributes a one-dimensional character factor. The remaining $(e_1, e_2)$-action carries the flux and therefore must realize a $q$-dimensional Weyl pair, which is exactly $\rho_\kappa^{(12)}$. Irreducibility then forces the tensor product form (standard for direct products with commuting factors). $\square$

This example illustrates a common geometry: holonomy may live in a lower-dimensional subspace (a set of flux planes), while other directions remain untwisted and contribute ordinary Fourier frequencies.

## I.3. General block sizes from the radical of the commutator bicharacter

The constant-flux case yields uniform block size $q$. More generally, a cocycle class is characterized by its commutator bicharacter $\beta_\sigma$ (Appendix B.1), and the block size is controlled by the radical of $\beta_\sigma$.

**Definition I.3** (Radical of an alternating bicharacter). Let $\beta : G \times G \to U(1)$ be a bicharacter. Its *radical* is the subgroup

$$\mathrm{Rad}(\beta) \triangleq \{g \in G : \beta(g, h) = 1 \ \forall h \in G\}.$$

**Proposition I.4** (Uniform block size and number of blocks for abelian $G$). *Let $G$ be finite abelian and fix a cocycle class with commutator bicharacter $\beta$. Then the twisted group algebra decomposes as*

$$\mathbb{C}_\sigma[G] \cong \bigoplus_{m=1}^{|\mathrm{Rad}(\beta)|} \mathrm{Mat}_d(\mathbb{C}), \qquad d = \sqrt{\frac{|G|}{|\mathrm{Rad}(\beta)|}}.$$

*Equivalently, all irreducible $\sigma$-projective representations have the same dimension $d$, and there are $|\mathrm{Rad}(\beta)|$ inequivalent irreps.*

*Proof sketch.* Elements $\{u_r : r \in \mathrm{Rad}(\beta)\}$ commute with all $u_g$ and hence span a commutative subalgebra of the center. For abelian $G$, the center of $\mathbb{C}_\sigma[G]$ is exactly the span of these elements, so $\dim Z(\mathbb{C}_\sigma[G]) = |\mathrm{Rad}(\beta)|$. Since $\mathbb{C}_\sigma[G]$ is finite-dimensional semisimple, it decomposes into a direct sum of simple matrix algebras, and the center dimension equals the number of simple components. Finally, the total dimension is $|G|$, so if there are $m = |\mathrm{Rad}(\beta)|$ components each of size $d^2$, then $md^2 = |G|$, giving $d = \sqrt{|G|/m}$. A complete proof can be given by constructing irreps via induction from maximal isotropic subgroups of $G/\mathrm{Rad}(\beta)$. $\square$

**Recovering constant flux.** In the 2D flux-$p/q$ case, $\mathrm{Rad}(\beta)$ consists of displacements whose first two coordinates are multiples of $q$, hence $|\mathrm{Rad}(\beta)| = (N/q)^2 = L^2$. Proposition I.4 then yields $d = \sqrt{N^2/L^2} = q$, matching Appendix E.

## I.4. Multiple independent flux planes and tensor-product blocks

Finally, we record an explicit higher-dimensional construction that decomposes holonomy into independent "flux planes."

Let $G = \mathbb{Z}_N^{2r} \times \mathbb{Z}_N^{d-2r}$ and let $\beta$ be an alternating bicharacter that, after a change of basis of $G$, factors into $r$ independent Weyl commutation relations: $U_{e_{2j-1}}U_{e_{2j}} = \omega_j U_{e_{2j}}U_{e_{2j-1}}$, $j = 1, \ldots, r$, with all other pairs commuting, where each $\omega_j$

is a primitive $q_j$th root of unity (with $q_j \mid N$). Then an explicit irreducible representation is the tensor product of the $r$ corresponding $q_j$-dimensional shift/clock representations (Appendix C.6) together with ordinary characters on the remaining commuting directions. Consequently, the block size is $d = \prod_{j=1}^{r} q_j$, and the number of blocks is $|G|/d^2$, consistent with Proposition I.4. This example is useful as a "normal form" intuition: holonomy introduces a finite number of independent noncommuting planes, each contributing a factor $q_j$ to the intrinsic representation dimension.

**Remark on canonical forms.** A full classification of alternating bicharacters on $\mathbb{Z}_N^d$ can be obtained via an integer change-of-basis bringing the bicharacter matrix to a symplectic Smith normal form, yielding exactly such independent planes. This aligns with the classification viewpoint in Appendix B but avoids committing to a specific cocycle representative.

# J. Extended Related Work Notes

This appendix provides additional context beyond the main related-work discussion, emphasizing connections that clarify the scope and novelty of holonomy-aware spectral prediction.

## J.1. Successor representations, predictive maps, and transfer

The successor representation (SR) was introduced in reinforcement learning as a way to factor long-horizon value estimation into dynamics-dependent prediction and reward-dependent readout (Dayan, 1993). The SR-as-resolvent viewpoint also underlies work linking predictive representations to hippocampal and entorhinal computations (Stachenfeld et al., 2017; Gershman, 2018). In RL, related decompositions appear in successor features and policy transfer frameworks (e.g., (Barreto et al., 2017)). Our contribution differs in that the central object is not a particular SR matrix but the *action-indexed family* of directed transition operators, and the key invariant controlling reuse is geometric (holonomy) rather than purely algebraic (commutativity).

## J.2. Spectral state representations and diffusion operators

Spectral representations for MDPs have a long history, including proto-value functions and Laplacian-based bases (Mahadevan & Maggioni, 2007), which connect to Laplacian eigenmaps (Belkin & Niyogi, 2003) and diffusion maps (Coifman & Maggioni, 2006). These approaches typically emphasize *undirected* or reversible diffusion structure. When directionality is present, one can either symmetrize (losing directional information) or work with non-self-adjoint operators.

A particularly close geometric cousin to our setup is the *connection Laplacian* viewpoint in manifold learning: vector diffusion maps (VDM) (Singer & Wu, 2012) generalize diffusion maps by transporting vectors (or more general fiber data) along edges before averaging. This introduces a gauge freedom and a holonomy-like notion through path-dependent transport. Our framework can be viewed as a controlled/MDP analogue: actions induce structured parallel transport (via the projective translation algebra), and holonomy classes control whether a single reusable spectral code exists (Section 6).

Related synchronization problems also exploit complex phases to encode cycle structure; angular synchronization (Singer, 2011) is a representative example in which gauge-invariant phase consistency is extracted from a Hermitian matrix built from pairwise offsets.

## J.3. Directed actions, grid-like codes, and translation structure

The directed-action generalization problem studied by Yu et al. (Yu et al., 2021) identifies a compelling mechanism for basis reuse when actions act as (commuting) translations: a shared Fourier-like eigenbasis supports prediction over different directed actions with action-dependent eigenvalues. This mechanism dovetails with the broader empirical and theoretical literature on grid-like representations in neural systems and learned agents (Cueva & Wei, 2018; Banino et al., 2018; Whittington et al., 2020). Our work can be read as characterizing *exactly* when the Yu-style reuse mechanism can and cannot extend: holonomy triviality is the precise condition under which scalar Fourier reuse is possible, and nontrivial holonomy forces a move to matrix-valued spectral codes.

## J.4. Directed graph Laplacians, magnetic phases, and complex embeddings

There is a substantial literature on spectral methods for directed graphs and nonreversible Markov chains, including directed Laplacians and associated Cheeger-type inequalities (Chung, 2005). More recently, *magnetic* deformations of graph

Laplacians attach complex phases to edges to encode directionality and cyclic flow (Fanuel et al., 2018), and these ideas have been used as inductive bias in directed-graph neural architectures (Zhang et al., 2021). These works share the theme that *cycle-consistent phases* are fundamental.

Our setting differs in two key ways. First, we study a *family* of controlled transition operators and long-horizon prediction via the SR resolvent, rather than a single embedding operator. Second, the phases arise from an *action geometry* (a projective translation structure), so the relevant invariant is not an arbitrary assignment of edge phases but a cohomology class that governs simultaneous block diagonalization across actions (Theorem 5.2 and Theorem 6.2).

### J.5. Gauge symmetry in geometric deep learning

The role of gauge freedom and local frame choices has become increasingly explicit in geometric deep learning (Bronstein et al., 2021). Group-equivariant convolutional networks (Cohen & Welling, 2016) generalize translation equivariance by building representations induced by group actions. Beyond global symmetries, gauge-equivariant convolutional networks (Cohen et al., 2019) formalize equivariance to local changes of reference frames, closely mirroring the mathematical language of bundles, connections, and holonomy.

Our work is not a network construction, but the conceptual link is strong: holonomy is the obstruction to defining globally consistent "aligned" features under transport, and gauge transformations capture the non-identifiability of local phases/frames. The main difference is that in our controlled-MDP setting, the transport operators are action-indexed and the central question is *predictive transfer across directed actions*, leading to the holonomy-class transfer theorems of Section 6.

### J.6. Classical magnetic translation symmetry

Finally, Appendix E makes explicit that the constant-flux holonomy model coincides algebraically with the magnetic translation group on a lattice (Zak, 1964). The $q \times q$ block matrices appearing in (44) are discrete Harper operators (Harper, 1955) whose spectrum as a function of rational flux produces the Hofstadter butterfly (Hofstadter, 1976). We emphasize that our goal is not to analyze the butterfly itself, but to import the representation-theoretic lesson into RL prediction: *nontrivial curvature forces matrix-valued spectra*, and these blocks are the correct reusable units for generalization across directed actions.

