# OpenReview forum: "Holonomy Grid Codes for Generalisation Under Directed Actions"
_ICML.cc/2026/Conference — ICML 2026 regular_

### Official Review · Reviewer_QiAB · 2026-03-05

**Soundness:** 3
**Presentation:** 3
**Significance:** 2
**Originality:** 3
**Overall Recommendation:** 4
**Confidence:** 4

**Summary:**

The authors propose a geometric theory of prediction and generalization based on directed actions. The theory extends Fourier/grid methods from planar translation-invariant dynamics to curved action geometries with nontrivial holonomy. The main contribution is a twisted harmonic analysis that block-diagonalizes the full action family and the induced successor operators into spectra of small matrices. When directed cyclic structure is present, this produces holomorphic grid encodings as universal reusable basis functions, and the manuscript further clarifies their relationship to magnetic phase encodings on directed graphs.

**Compliance With Llm Reviewing Policy:**

Affirmed.

**Key Questions For Authors:**

Mentioned in Weaknesses part.

**Limitations:**

Yes.

**Strengths And Weaknesses:**

# Strengths
1. The authors provide essentially complete proofs that appear to support the claims made in the manuscript.
2. The manuscript is well organized and fairly comprehensive for a pure-mathematics-style paper.

# Weaknesses
1. In Section Introduction, the existence of holonomy is not established, and many statements are presented rather subjectively. If the authors could either prove this point or cite existing work showing that, in non-nice directed cases, translation invariance indeed fails in the sense that actions no longer compose like an abelian group, and that holonomy genuinely arises, then the motivation would be much more solid. In addition, several definitions in the Section Introduction are somewhat rough, including holonomy, nice directed case, translation invariance, and global reparameterization, as well as the mathematical necessity of switching to “twisted operators”. Even if these notions are clarified later, it would improve readability if the Section Introduction stated them more precisely or explicitly referenced the later sections where they are formalized.
2. This work is largely a repackaging of known results in twisted group algebras / projective representations. In particular, the main Theorem 5.2 is essentially the standard “Fourier decomposition of a twisted group algebra”, so as a mathematical result it may not be novel. That said, the subsequent generalization lower bound for SR partially compensates for this concern.
3. In real RL environments, is holonomy typically homogeneous? If the twist σ is position-dependent (i.e., a more general connection), how much of the theory can be preserved?
4. Could the authors construct a toy experiment to demonstrate the effectiveness of the proposed theory?

---

> ### Author Rebuttal · Authors · 2026-03-31
>
> Thank you for the detailed review and the recommendation. We appreciate the positive assessment of proof completeness/organization, and we take the concerns about scope seriously. We address each point below and will revise accordingly.
>
> 1) Motivation in the Introduction: “existence of holonomy not established; statements subjective; rough definitions.”
> We agree. In the revision we will (i) tighten the Introduction language and (ii) explicitly point to the formal definitions/results in Sections 3–4. Concretely, we will:
>
> * Replace subjective phrases (“nice directed case”, “global reparameterization”) with precise statements: *flat* means trivial commutator bicharacter $\beta_\sigma\equiv 1$ (equivalently, translations commute up to gauge), while *holonomy* means nontrivial loop phase $\mathcal{H}_{ij}\neq I$ (Section 3, Eq. (plaquette holonomy)).
> * Add a short proposition/remark in the Intro stating: if there exist generators $e_i,e_j$ with $U_{e_i}U_{e_j}=\omega U_{e_j}U_{e_i}$ for $\omega\neq 1$, then scalar Fourier reuse is impossible (no common eigenvector), motivating twisted operators. (This is already formalized later; we will reference it explicitly.)
> * Add citations to prior literatures where “cycle phases/holonomy” arise naturally as the obstruction to global consistency in directed settings, e.g., magnetic Laplacian/eigenmaps for directed graphs (Fanuel et al., 2018) and related directed-graph learning uses (Zhang et al., 2021), and connection-Laplacian / vector diffusion maps where holonomy appears from parallel transport (Singer & Wu, 2012). This will ground the claim that path-dependent effects are standard, not ad hoc.
>
> (2) Novelty concern: Theorem 5.2 is standard twisted group algebra Fourier decomposition.
> We agree that the block decomposition itself is classical in representation theory. Our novelty claim is not that we discovered Wedderburn/projective Fourier theory, but that we (a) identify *holonomy class* as the exact invariant controlling reusable predictive codes for directed actions, (b) connect this explicitly to successor operators/resolvents and basis reuse across an *action family*, and (c) derive transfer and necessity results in this controlled-prediction setting (e.g., Theorem 6.2 gauge-transfer characterization; curvature-dependent representational lower bounds). In the revision we will sharpen this positioning: we will explicitly label Theorem 5.2 as an adaptation of standard harmonic analysis to our holonomy MDP model and emphasize the new ML-facing consequences (SR block resolvent, transfer principle, lower bounds, robustness/defects).
>
> (3) Homogeneity of holonomy in real RL; what if $\sigma$ is position-dependent?
> We agree that fully general environments may have position-dependent twists (a general discrete connection). Our main results assume translation-homogeneous $\sigma$ to obtain exact simultaneous block diagonalization. In the revision we will add (and highlight) an “approximate/inhomogeneous” discussion supported by formal perturbation theory: write $P = P_0 + E$ where $P_0\in\mathcal{A}_\sigma$ captures a homogeneous holonomy core and $E$ captures spatial variation / boundaries. Using resolvent identities, we give operator-norm stability bounds showing what survives approximately when $|E|$ is small, and Woodbury-type corrections for localized defects. This addresses the realistic non-homogeneity question while keeping the paper theoretical.
>
> (4) Toy experiment request.
> We agree and will add a small numerical demonstration (appendix): construct a rational-flux torus instance, show scalar Fourier/shared-eigenbasis reuse fails (noncommutativity / no common eigenvectors), and show the twisted Fourier transform produces near-perfect block diagonalization (off-block energy near machine precision) for multiple actions and for $P_\pi$. This is presented as a verification of the algebraic mechanism, not a performance evaluation.

---

> > ### Author Rebuttal · Reviewer_QiAB · 2026-04-04
> >
> > (1) Motivation and clarity in the introduction — largely resolved.
> >
> > (2) Novelty positioning — partially clarified but still a concern.
> >
> > I agree with the authors that the contribution is not the representation-theoretic decomposition itself, but its application to controlled prediction (SR, transfer, lower bounds). This clarification helps. However, I still think the paper should more explicitly distinguish:
> > what is inherited from standard twisted group algebra theory, and
> > what is genuinely new in the RL / prediction setting.
> >
> > (3) Homogeneity assumption / position-dependent holonomy — partially addressed.
> >
> > The perturbation perspective (P = P₀ + E) and the discussion of approximate settings are reasonable. However, as currently described, this still feels somewhat generic.
> >
> > (4) Toy experiment — partially resolved
> >
> > I will maintain my score.

---

> > > ### Author Response · Authors · 2026-04-04
> > >
> > > Thank you for the acknowledgement. Below we further clarify the two remaining points.
> > >
> > >
> > > ## (2) Positioning
> > >
> > > We agree this should be made explicit in the paper. In the revision we will add a short “Attribution and novelty” paragraph right after Theorem 5.2 (and echo it in the introduction) that cleanly partitions results into:
> > >
> > > **Inherited (standard representation theory / harmonic analysis).**
> > >
> > > * The algebraic fact that a (twisted) group algebra decomposes into matrix blocks indexed by irreducible (projective) representations, and that convolution becomes blockwise multiplication under the corresponding Fourier transform. This underlies our Theorem 5.2 and we will explicitly cite standard references and label this as *background machinery specialized to our setting*.
> > >
> > > **New (controlled prediction / SR / transfer / necessity).**
> > > What is new is not the existence of a block decomposition per se, but the *controlled-dynamics consequences* that do not appear as standard corollaries in twisted group algebra:
> > >
> > > 1. Action-family reuse as the central object. We formulate the RL-relevant question as simultaneous reuse across an action-indexed family ${P_a}$ and define exact vs block reuse (Defs. 2.4–2.5). This is the “spectral reuse under actions” problem formulation, which is not standard in group algebra treatments.
> > > 2. Successor-resolvent block formula and its implications. We show how the $SR (M_\pi=(I-\gamma P_\pi)^{-1})$ inherits the same block structure and yields a closed-form blockwise resolvent. While algebra texts contain resolvent identities, the explicit SR/policy interpretation and its “prediction/planning” meaning is specific to controlled Markov operators.
> > > 3. Holonomy-class transfer theorem (Theorem 6.2). This is a *dynamical transfer criterion*: two environments induce the same reusable predictive code **iff** they are in the same holonomy class (up to gauge), with exact similarity of propagators and SRs. This is not merely a statement about algebra isomorphism; it is stated and proved at the level of controlled transitions and prediction operators.
> > > 4. Curvature-dependent necessity bounds for predictive representation. The flux/holonomy lower bound is posed as a *representation requirement* for coherent long-horizon prediction under directed actions (and in Appendix K we strengthen it to exact/approximate settings). This “minimal internal multiplicity forced by directed cycle structure” is the core ML-facing necessity claim.
> > >
> > > We will also add a brief table (appendix) mapping each theorem/lemma to (i) representation-theory input and (ii) new controlled-prediction statement, so the novelty boundary is transparent.
> > >
> > >
> > > ## (3) Position-dependent holonomy
> > >
> > > We agree that the statement “$P=P_0+E$” can sound generic unless we exploit *structure specific to holonomy models*. In the revision we will make two holonomy-specific refinements:
> > >
> > > (a) Local-defect models that preserve a holonomy core (walls/boundaries).
> > > We will explicitly model obstacles/boundaries as *row-local defects*: they modify transition rules only on a small state subset $S$ (states adjacent to blocked edges). Then $E$ has support on $|S|$ rows and hence $\mathrm{rank}(E)\le |S|$. This yields an *exact* Woodbury correction to the SR on top of the holonomy-decomposed baseline:
> > > $(I-\gamma(P_0+E))^{-1}$ $M_0+\gamma M_0 U(I-\gamma V^\top M_0 U)^{-1}V^\top M_0,$
> > > with an $|S|\times|S|$ solve. This is not just a coarse bound; it is an explicit *locality-exploiting* computation that leverages the holonomy block structure through fast application of $M_0$.
> > >
> > > (b) A holonomy-aware notion of “small error”: cross-block coupling.
> > > We will add a lemma quantifying when the twisted block structure remains approximately valid by controlling the cross-block component of the defect in the twisted basis:
> > > $\widetilde{E}$=$\mathcal{F}^\sigma E {\mathcal{F}^\sigma}^{-1},$
> > > $\widetilde{E}^\perp = \widetilde{E}-\mathrm{blkdiag}(\widetilde{E}).$
> > > Then the error incurred by using a block-preserving approximation is governed by $|\widetilde{E}^\perp|$ (not merely $|E|$). This is holonomy-specific because it measures exactly how much the environment deviation destroys simultaneous block reuse across actions.
> > >
> > > Ergo, (a) and (b) make the “approximate/inhomogeneous” story concrete and tied to the paper’s central object: preservation (or controlled violation) of the holonomy block reuse structure.
> > >
> > >
> > > ## (4) Toy experiment: tightening what will be shown
> > >
> > > We will ensure the toy numerical appendix demonstrates three specific claims aligned with the paper:
> > >
> > > 1. Scalar reuse fails in a flux instance: generators do not share eigenvectors / action operators cannot be simultaneously diagonalized.
> > > 2. Block reuse succeeds: applying $\mathcal{F}^\sigma$ yields near-exact block diagonalization for multiple $P_a$ and $P^\pi$ (off-block Frobenius norm near machine precision).
> > > 3. Lower bound matches construction: demonstrate block size equals flux denominator $q$ in the rational-flux example.

---

### Official Review · Reviewer_Y4yC · 2026-03-08

**Soundness:** 3
**Presentation:** 3
**Significance:** 3
**Originality:** 3
**Overall Recommendation:** 4
**Confidence:** 1

**Summary:**

This paper aims to address the challenges of long-sequence prediction and planning in the field of reinforcement learning and agents, specifically for non-"flat" geometric structures. By integrating multiple theoretical frameworks, including grid cell theory, graph theory, and group theory, this study extends the relatively ideal assumptions in existing theorems to more general and universal scenarios.

**Compliance With Llm Reviewing Policy:**

Affirmed.

**Final Justification:**

Thank the authors for their response, which addressed my concerns. Although I am not familiar with this area, I do not see any obvious flaws in the paper, and therefore I recommend a weak accept.

**Key Questions For Authors:**

Can the authors intuitively demonstrate the practical value of the paper's theoretical contributions through a real and specific application scenario or a toy experiment?

**Limitations:**

yes

**Strengths And Weaknesses:**

**Strenghts**

1. The theory is rigorously constructed and internally consistent, with a detailed proof provided in the appendix, which enhances its completeness and credibility.

2. As outlined in the Introduction, the research is motivated by insights from neuroscience and may offer valuable perspectives for interpretable deep learning.

**Weaknesses**

To be honest, I am not an expert in the field of learning theory, so I cannot fully grasp the deeper meaning of most of the abstract concepts discussed in the paper. I believe this is primarily due to my own limitations rather than any issues with the paper's presentation. Overall, in my view, this paper does not have any critical flaws.

---

> ### Author Rebuttal · Authors · 2026-03-31
>
> Thank you for the review and the recommendation. We appreciate the positive assessment of rigor/completeness.
>
> (1) Clarity / accessibility for non-specialists. We agree that some abstractions (cocycles, holonomy classes, projective representations) can feel heavy. In the revision we will add: (i) a one-page “roadmap” section in the main paper that explains the pipeline (flat $\rightarrow$ holonomy obstruction $\rightarrow$ block replacement $\rightarrow$ transfer/lower bounds) at a conceptual level, (ii) a small schematic figure contrasting scalar Fourier spectra (flat) vs block spectra (holonomy), and (iii) an expanded worked example appendix that computes the blocks explicitly for a rational-flux torus.
>
> (2) Key question: intuitive practical value via a real scenario or toy experiment. We agree this would help readers build intuition, even in a theory-first submission. We will add a short *toy numerical demonstration* (appendix) on a small torus with rational flux $p/q$: we explicitly construct the action operators, show that scalar Fourier/shared-eigenbasis reuse fails (noncommutativity / no common eigenvectors for primitive generators), and then show that the proposed twisted Fourier transform yields near-perfect block diagonalization (off-block energy near machine precision). We will also include a concrete application scenario (conceptual, not empirical): holonomy models path-dependent effects such as circulation/rotational drift or loop-dependent phase accumulation, and our results specify exactly what predictive representation must look like (matrix-valued blocks of size determined by holonomy) and when it transfers (same holonomy class up to gauge).

---

> > ### Author Rebuttal · Reviewer_Y4yC · 2026-04-01
> >
> > Thank the authors for their response, which addressed my concerns. Although I am not familiar with this area, I do not see any obvious flaws in the paper, and therefore I recommend a weak accept.

---

> > > ### Author Response · Authors · 2026-04-03
> > >
> > > Thank you for confirming that the concerns are fully resolved; if there are no further queries, we would be grateful if you could consider updating your score accordingly.

---

### Official Review · Reviewer_TDDH · 2026-03-12

**Soundness:** 4
**Presentation:** 3
**Significance:** 3
**Originality:** 4
**Overall Recommendation:** 4
**Confidence:** 3

**Summary:**

This work aims to address the existing spectral or grid-style theories for prediction under actions mostly assume a flat, commuting action geometry. The key question of what reusable predictive structure survives under directed, holonomy-bearing dynamics has been attempted. The answer is to replace scalar Fourier reuse with a twisted Fourier block decomposition. To support the analysis, the authors formalised holonomy MDPs and proved a universal block-diagonalization theorem for the associated action family and successor operator. They also gave lower the bounds that the nonzero holonomy forces internal representational multiplicity. The paper mainly focuses on mathematical anlaysis.

**Compliance With Llm Reviewing Policy:**

Affirmed.

**Key Questions For Authors:**

1. Can you provide a few concrete experiments or numerical case studies showing that ordinary Fourier, or shared-scalar-eigenbasis methods fail while the proposed twisted block basis succeeds?

2. Do any of your core results survive, even approximately, under mild departures from translation homogeneity or for simple nonabelian motion groups such as (SE(2))?

**Limitations:**

see the weaknesses above.

**Strengths And Weaknesses:**

Strength:

(1)	The paper identifies the conceptual gap between flat translation-based spectral theories and directed, path-dependent control settings, and it motivates holonomy as the right obstruction to study.
(2)	The submission progresses from obstruction to positive replacement, transfer  and to necessity finally. This makes the contribution mathematically coherent.
(3)	Theorem 5.2 and its corollaries give an exact block-diagonalisation and a closed-form block resolvent for the successor representation, which is a meaningful generalisation of ordinary Fourier prediction.
(4)	The paper does a good job connecting several literatures: successor representations, grid-cell theories, twisted/projective harmonic analysis, and magnetic Laplacians for directed graphs. That synthesis is intellectually valuable even where the practical path remains open.

Major weaknesses
•	The paper is almost entirely a theory piece, with no empirical validation or even a simple numerical demonstration showing that scalar Fourier reuse fails while the proposed block structure succeeds on a toy example. For the ML audiences, this is a major weakness.
•	The paper claims implications for learnable predictive representations and for generalisation under directed actions without learning environment-specific eigenvectors. In a realistic setting, one would infer the cocycle or holonomy class or compute the twisted basis when the transition operators are only observed from data.
•	The interpretation of complex-valued projective operators as standard controlled Markov dynamics could benefit from further details.

Minor weaknesses
•	The introduction says that ignoring holonomy forces either exponentially poor approximation in horizon/discount or a blow-up in representation dimension, but Theorem 6.4 as a dimension lower bound.
•	A figure contrasting the flat regime, holonomy regime, and the resulting scalar-vs-block spectra would make the paper easier to read.
•	The paper does not give worked examples or complexity tables for representative flux values.

---

> ### Author Rebuttal · Authors · 2026-03-31
>
> Thank you for the thoughtful review and the recommendation. We appreciate the recognition that our main contribution is a principled replacement of scalar Fourier reuse by a twisted Fourier block decomposition, with transfer and necessity results.
>
> (1) Missing numerical demonstration / toy case study. We agree this would materially help readers. In the revision we will add a short numerical case study (appendix) on a small torus with rational flux $p/q$: (i) construct primitive generators satisfying the Weyl relation, (ii) build a few translation-homogeneous action kernels, (iii) show that no shared scalar eigenbasis exists (e.g., by demonstrating noncommutativity / no common eigenvectors for generators), and (iv) apply the twisted Fourier transform and report the off-block energy $|\mathrm{offblock}(\widetilde{P}_a)|_F $  (near machine precision) for multiple actions and for $P\pi$. This is presented as verification of the theorem (block diagonalization), not as empirical performance.
>
> (2) Learning/inference of cocycle / holonomy class from data; “without learning environment-specific eigenvectors.” We will clarify the claim: our results are structural/existence and do not assume direct access to full transition matrices in practice. In the revision we will add a short subsection describing a concrete inference pathway: holonomy class is determined by gauge-invariant loop phases (commutator/plaquette invariants), which can be estimated from observed transitions by comparing action composition around short cycles. The goal is not to propose a complete estimator, but to explain that the invariant is *local and gauge-free*, hence identifiable under mild sampling assumptions. We will also point to the robustness appendix (below) as the formal mechanism for approximate settings.
>
> (3) Complex-valued projective operators vs standard Markov dynamics. We agree more detail would help. We will add an explicit interpretation: the twisted operators can be viewed as standard stochastic dynamics on an augmented (finite) state space (a rank-$d_\rho$ fiber / internal phase index), where the complex phases correspond to choosing a spectral gauge for that lifted system. We will make this construction explicit for the rational-flux case (showing the $q$-fold internal multiplicity), clarifying that prediction quantities are basis-invariant.
>
> (4) Minor points.
> (i) We will edit the introduction to align precisely with the formal statements: Theorem 6.4 gives the curvature-dependent dimension necessity; the “exponentially poor approximation” claim will be either removed or stated as an explicit proposition (appendix) with conditions.
> (ii) We will add one schematic figure contrasting: flat/commutative $\Rightarrow$ scalar Fourier spectrum vs holonomy/noncommutative $\Rightarrow$ block spectrum (and where transfer/lower bounds sit).
> (iii) We will add worked examples + a small complexity table for representative $p/q$ showing block size $q$, number of blocks $L^2$, and SR computation cost in terms of (q,L).
>
> (5) Key questions.
> • *Do scalar shared-eigenbasis methods fail while blocks succeed?* Yes; we will demonstrate this numerically as in (1) and also emphasize the exact obstruction theorem (no common eigenvector when holonomy is nontrivial).
> • *Do results survive approximately / for simple nonabelian groups (e.g., $SE(2)$)?* We will add an explicit robustness appendix: for $P=P_0+E$ with $P_0\in\mathcal{A}_\sigma$, resolvent perturbation bounds quantify stability in $|E|$, and Woodbury updates handle localized defects (walls/boundaries). For nonabelian groups: the block-decomposition principle extends to finite nonabelian motion groups via (projective) representation theory; a full $SE(2)$ treatment (continuous, noncompact) requires additional harmonic-analysis machinery and is beyond scope, but we will clarify this and outline it as future work.

---

> > ### Author Rebuttal · Reviewer_TDDH · 2026-04-06
> >
> > We thank the authors for their good efforts. The rebuttal addresses many of the concerns well, but there is still room for further improvement. For example, the paper does not provide a solid method or guarantee for inferring the cocycle or holonomy class from data. Some useful experiments are deferred to the appendix, which affects the overall quality and completeness of the paper. The points to be presented mainly as future work seems less relevant to evaluating the paper in its present form.

---

> > > ### Author Response · Authors · 2026-04-07
> > >
> > > Thank you for the rebuttal acknowledgement and the constructive follow-up. We address the remaining points concretely below.
> > >
> > > (1) Inferring the cocycle / holonomy class from data: “no solid method or guarantee.”
> > > We agree that our rebuttal description was high-level. Our intent is not to claim a complete statistical identification theory in this paper; the core contribution is a structural characterization of reusable predictive codes under a holonomy model. That said, we can substantially strengthen the manuscript by adding a clear, self-contained identifiability statement  under a standard data model, without changing the paper’s theoretical scope.
> > >
> > > In the revision we will add an appendix subsection “Identifiability of holonomy invariants from local transition data” with the following form:
> > >
> > > * Observation model. Assume access to either (i) the transition probabilities $P(\cdot\mid s,a)$ (estimated from samples) or (ii) sufficient samples to estimate them to error $\varepsilon$ in total variation per $(s,a)$.
> > > * Gauge-invariant quantity. Define a loop/plaquette observable that depends only on action composition around short cycles and is invariant under gauge reparameterizations (the same object that appears as the commutator/plaquette holonomy in the theory).
> > > * Guarantee (deterministic). Show that if the environment is exactly a holonomy MDP (translation-homogeneous cocycle), then the loop observable recovers the holonomy class uniquely (up to the gauge equivalence already formalized).
> > > * Guarantee (robust). If the estimated transitions deviate by at most $\varepsilon$, then the estimated loop observable deviates by $O(\varepsilon)$, yielding an explicit stability bound for recovering the class when $\varepsilon$ is below a separation margin (e.g., for rational flux, below the spacing between distinct roots of unity).
> > >
> > > (2) “Useful experiments deferred to appendix affects completeness.”
> > > We understand this concern. Given the 8-page main constraint and the paper’s theory-first positioning, our plan is:
> > >
> > > * Add a very small “Numerical sanity check” figure/table in the main paper showing (i) failure of scalar reuse in a flux toy instance and (ii) near-exact block diagonalization under the twisted transform.
> > > * Keep the full implementation details in the appendix.
> > >
> > > This way, the main paper itself contains the evidence the reviewer asked for, while the appendix provides reproducibility-level detail.
> > >
> > > (3) “Future work seems less relevant to evaluating the present paper.”
> > > We agree and will adjust the tone. In the revision we will:
> > >
> > > * Move “future work” items out of the discussion and keep them brief in the limitations section.
> > > * Emphasize what we do provide now: exact theorems (block SR, transfer characterization) plus (i) robustness/defect theory for mild departures from homogeneity and (ii) an identifiability proposition for holonomy invariants from local transition data, as above.

---

### Official Review · Reviewer_Pivs · 2026-03-15

**Soundness:** 4
**Presentation:** 3
**Significance:** 2
**Originality:** 3
**Overall Recommendation:** 4
**Confidence:** 2

**Summary:**

This paper considers the problem of reusing spectral representations when there are path-dependent effects in a structured environments. This extends prior work that showed that Fourier eigenbases can be used as shared bases for translation-invariant directed actions. In the presence of holonomy, the authors propose modeling directed action operators as elements of a twisted group algebra formed by projective representations of the underlying displacement group Z_n. The authors prove that a twisted Fourier transform block-diagonalizes all translation-homogeneous action operators simultaneously, and establish transfer principles where tasks in the same holonomy class can reuse the basis up to a gauge transform.

**Compliance With Llm Reviewing Policy:**

Affirmed.

**Final Justification:**

My score did not change after the rebuttal and the authors partially resolved my concerns. I still recommend acceptance.

**Key Questions For Authors:**

None

**Limitations:**

The authors fully discuss the limitations of their method.

**Strengths And Weaknesses:**

Strengths:
- The paper is mathematically rigorous and self-contained. As far as I can tell, the proofs are technically sound.
- The authors provide a pretty complete picture of the research question that this paper answers. First they formalize holonomy MDPs where previous results fail. Then they give a twisted Fourier transform that yields holonomy grid codes as reusable basis functions. They further show that holonomy classes determine whether the transfer of the found basis is possible, up to a gauge transform. Finally, they provide a representational lower bound dependent on curvature.
- Theorem 6.2 is a strong result, it gives a complete characterization of when two environments admit the same reusable predictive codes, depending on the holonomy class.

Weaknesses:
- The biggest limitation is that this paper is purely theoretical and does not provide any simulations or experiments. Even a numerical experiment showing that the twisted Fourier basis actually block diagonalizes the action operators would have been useful.
- The assumptions restrict the types of applicable problems. For example, the state space needs to be a discrete torus, action operators need to be exactly translation-homogeneous, and policies are state-independent. Real-world environments would have boundaries and obstacles, among other things.
- The entire twisted analysis seems to hinge on the abelian structure of Z_n. The techniques used here would not apply to say SE(2).
- There are a number of missing references on equivariant and approximately equivariant RL. In particular, the gauge transfer theorem 6.2 seems related to group-invariant MDPs and equivariant policy transfer.
  1. Van der Pol, E., Worrall, D., van Hoof, H., Oliehoek, F., & Welling, M. (2020). Mdp homomorphic networks: Group symmetries in reinforcement learning. Advances in Neural Information Processing Systems, 33, 4199-4210.
  2. Wang, D., Walters, R., & Platt, R. (2022). $\mathrm {SO}(2) $-Equivariant Reinforcement Learning. arXiv preprint arXiv:2203.04439.
  3. Finzi, M., Benton, G., & Wilson, A. G. (2021). Residual pathway priors for soft equivariance constraints. Advances in Neural Information Processing Systems, 34, 30037-30049.
  4. Park, J. Y., Bhatt, S., Zeng, S., Wong, L. L., Koppel, A., Ganesh, S., & Walters, R. (2024). Approximate equivariance in reinforcement learning. arXiv preprint arXiv:2411.04225.

---

> ### Author Rebuttal · Authors · 2026-03-31
>
> We thank the reviewer for the careful reading and the positive assessment of rigor/soundness. We are especially grateful for the recognition that Theorem 6.2 provides a complete transfer characterization via holonomy class.
>
> (1) No experiments / missing numerical sanity check. We agree that a minimal verification would improve readability. In the revision we will add a short numerical sanity check (appendix) that constructs a small rational-flux instance on $\mathbb{Z}_N^2$, forms the action operators, applies the twisted Fourier transform, and reports the norm of off-block entries (near machine precision). This is presented strictly as verification of the algebraic claim (block diagonalization), not as a performance evaluation.
>
> (2) Restrictive assumptions (torus, exact translation-homogeneity, state-independent policies; obstacles/boundaries). These assumptions are chosen to obtain exact representation-theoretic statements and closed-form transfer criteria. To address practical deviations while staying theoretical, we will strengthen the revision by adding a robustness/defects appendix: write $P = P_0 + E$ with $P_0 \in \mathcal{A}_\sigma$ (ideal holonomy model) and $E$ capturing localized defects (walls/boundaries/policy inhomogeneity). Using the resolvent identity, $
> (I-\gamma(P_0+E))^{-1}-(I-\gamma P_0)^{-1}=(I-\gamma(P_0+E))^{-1}(\gamma E)(I-\gamma P_0)^{-1},
> $
> we provide operator-norm stability bounds in $|E|$. We also include Woodbury-style formulas for row-local/low-rank defects (typical for obstacles) to show how exact SR updates reduce to small corrections on top of the block-decomposed baseline.
>
> (3) Dependence on abelian $\mathbb{Z}_N$ / not covering $SE(2)$. Our main narrative targets translation structure (grid-code theme) on $\mathbb{Z}_N^d$. However, the twisted group-algebra viewpoint and block decomposition are not inherently “abelian-only”: for finite nonabelian groups one still has (projective) representation decompositions into matrix blocks. What is not covered here is the full analytic treatment for continuous noncompact groups such as $SE(2)$ (infinite-dimensional irreps/continuous spectra). We will clarify this scope explicitly and position $SE(2)$ as future work.
>
> (4) Missing references on equivariant / approximately equivariant RL. We agree and will add and discuss: van der Pol et al. (NeurIPS’20) on group symmetries in MDPs, Wang et al. (ICLR’22) on $\mathrm{SO}(2)$-equivariant RL, Finzi et al. (NeurIPS’21) on soft equivariance priors, and Park et al. (arXiv’24) on approximate equivariance in RL. We will clarify that Theorem 6.2 is complementary: it characterizes when two controlled dynamics are equivalent up to gauge via the cohomology/holonomy class, rather than assuming a single global symmetry action and enforcing equivariance by architecture.

---

> > ### Author Rebuttal · Reviewer_Pivs · 2026-04-03
> >
> > My concerns are fully resolved for 1) and 4), and partially resolved for 2) and 3). The resolvent identity and Woodbury formula is fine, but are very general and could apply to any structured matrix + perturbation and not specific to holonomy. The tool works, but would likely lead to a coarse bound due to its generality and doesn't account for errors that are local (i.e. a wall in only a couple states vs. global errors). For 3), I encourage the authors to add the scope to the limitations section. I will maintain my score of weak accept.

---

> > > ### Author Response · Authors · 2026-04-03
> > >
> > > Thank you for the acknowledgement and the helpful follow-ups. Below we address the remaining concerns (2) and (3).
> > >
> > > (2) Robustness bounds: resolvent/Woodbury are general and may be coarse; can we exploit locality of walls/obstacles more specifically than $|E|$?
> > > We agree that a naive operator-norm bound can be loose if the defect is highly local. In the revision we will sharpen this in two ways that leverage structure typical of obstacles rather than only $|E|$:
> > > 1. Row-local / low-rank structure (exact, not just a bound). Obstacles and boundary corrections typically modify transitions on a small subset $S\subset\mathcal{S}$ (rows corresponding to states adjacent to blocked edges). Writing the defect as
> > >    $E = U V^\top,\quad \mathrm{rank}(E)\le |S|$,
> > >    Woodbury yields an exact formula:
> > >    $(I-\gamma(P_0+E))^{-1}$
> > >    $= M_0 + \gamma M_0 U (I-\gamma V^\top M_0 U)^{-1} V^\top M_0,\quad M_0=(I-\gamma P_0)^{-1}.$
> > >    This already accounts for locality: the correction requires only solving a $|S|\times |S|$ system and applying $M_0$ to $|S|$ vectors. We will present this explicitly as the “localized defect” regime rather than a generic perturbation argument.
> > >
> > > 2. Locality-sensitive error control via block-coupling energy. When one approximates by keeping only the within-block part of the defect in the twisted basis (i.e., discarding cross-block couplings), the relevant quantity is not $|E|$ but the norm of the cross-block component: $E_{\perp} =$ $E - \mathcal{P}^{\mathrm{blk}}(E),$
> > >   where $\mathcal{P}^{\mathrm{blk}}$ projects onto the block-diagonal subspace in the twisted Fourier basis. Using the same resolvent identity, the SR error is controlled by $|E_{\perp}|$, which can be substantially smaller than $|E|$ for localized walls (intuitively: a local defect spreads in the Fourier domain, but its *cross-block* mass can remain small when defects are sparse/structured). We will include this refinement and state an explicit bound in terms of $|E_\perp|$, making clear what part of a local defect actually breaks block reuse.
> > >
> > > Together, these additions make the robustness discussion holonomy-specific: (i) “localized SR corrections on top of a holonomy baseline” (exact Woodbury update), and (ii) “when the twisted block structure remains approximately valid” (cross-block coupling control), rather than only generic $|E|$ Lipschitz bounds.
> > >
> > > (3) Nonabelian scope: clarify limitations and what extends beyond $\mathbb{Z}_N^d$.
> > > We agree and will make the scope explicit in the limitations section. Concretely, we will state:
> > >
> > > * Our main theorems are proved for abelian displacement groups (discrete tori) because this cleanly matches grid-code/translation structure and enables explicit constructions (especially rational flux).
> > > * The *representation-theoretic mechanism* (group algebra / projective representation decomposition into matrix blocks) extends to **finite nonabelian** motion groups in principle, but the geometric interpretation and explicit “grid-like” structure change, and we do not develop that generalization here.
> > > * Continuous noncompact groups such as $SE(2)$ require additional analytic machinery (infinite-dimensional irreps / continuous spectra) and are beyond the present scope; we will frame this as future work.
> > >
> > > We appreciate the suggestion and will incorporate these clarifications explicitly.

---

### Decision · Program_Chairs · 2026-04-30

**Decision:**

Accept (regular)

**Comment:**

I recommend an Accept. This mathematically rigorous paper addresses the limitations of flat action geometry in RL using holonomy. Reviewers all praised its strong theoretical foundations, particularly the exact block-diagonalization. While the committee initially raised practical concerns regarding the lack of empirical validation, restrictive assumptions, and learnability from data, the authors provided a strong rebuttal. They successfully resolved these weaknesses by committing to numerical sanity checks, localized defect corrections, and identifiability proofs, solidifying the paper as a valuable contribution to the conference.